# SciPro Arena: a Case Study of AI Agent Capabilities in Scientific Analysis Tasks

## Abstract

In the physical sciences, stringent standards of error tolerance require data analysis to be either rigorous enough to be trusted by other scientists, or be completely disregarded. We introduce SciPro (Scientific Process) Arena, a benchmark that measures how reliably frontier AI systems analyze spectral scientific data. Using simulated condensed matter physics data, we prompt models to identify structures from 2D intensity arrays in the face of noise and limited instrumental resolution, scoring them such that a score of 1 indicates complete accuracy, while 0.5 denotes deviation by an extent equivalent to typical experimental error. We test recent reasoning models. With direct query, the best models score only 0.13 out of 1 averaged across all questions, rising to 0.20 for questions without noise. While enabling access to a Python interpreter improved scores to as much as 0.23 (averaged) and 0.39 (noiseless), these still pale in comparison to human–written code, which score 0.37 and 0.55 respectively. Clear failure modes were identified: models can find simple features, but struggle to trace continuous patterns or compute derived quantities. Performance predictably degrades with increased data resolution and noise levels. These results show current frontier models cannot reliably perform scientific data analysis, highlighting a significant gap between current capabilities and practical uses of LLMs for scientific discovery in physics.

## 1 Introduction

On the cusp of widespread adoption of AI systems in science, we need tests for rigorous, dependable scientific reasoning. Recent scientific benchmarks test competence in code and/or workflow generation, not whether models can accurately extract patterns from noisy experimental data, a core skill for scientific reasoning and autonomous agents. LLM training data includes published scientific figures, yet rarely includes the analysis process that created those figures. This process—extracting meaningful information from messy data—is what scientists and AI agents must do *reliably*. SciPro (Scientific Process) Arena fills this gap by testing models on accuracy in real spectral analysis tasks.

Condensed matter, the field responsible for developing much of modern technology—especially computing hardware that led to the advent of AI itself—promises scientific advances that will beget further breakthroughs in computing and other sciences (de Leon et al., 2021). Electronic structure is its '*genomic code*', telling us how electrons behave in materials, thereby explaining material properties. Condensed matter shares with AI the common thrust (Xiao et al., 2025) of studying extremely complex systems (on the order of Avogadro's constant, $\sim 10^{23}$), but lacks the generous breadth of analyzable features afforded to LLMs, *viz.* Golden Gate Claude (Templeton et al., 2024), because these features are accessible only if an experimental technique is physically feasible.

We focus on the tip of the spear of condensed matter, Angle–Resolved Photoemission Spectroscopy (ARPES) (Damascelli et al., 2003; Sobota et al., 2021), which measures electronic structure. ARPES was chosen for three strategic reasons:

1. Of the relatively limited number of techniques available, **ARPES stands out for its rich and more extended feature space**, as well as closeness to the ground truth (more direct linkage to physical models), relying on less theoretical scaffolding for its interpretation.

2. Domain–specific foundation models elsewhere in science have emphasized the **importance of learning the 'domain language' in which scientific data is naturally represented**

(Zhang et al., 2025), such as genomes in biology (Nguyen et al., 2024), and molecules in chemistry (Chithrananda et al., 2020; Kim et al., 2021). Embodying an understanding of electronic structure, the 'domain language' of condensed matter, is a necessary condition for learning an informative representation of material systems. Understanding electronic structure is crucial (Goyal et al., 2025) for **constructing foundation models for the critical field of novel materials, the reason for which condensed matter is all–important** (Trump, 2025). Because *grokking* ARPES data is the only way scientists currently have to reveal electronic structure (Yang et al., 2018), **the ability to process ARPES data is essential** to leveraging the capabilities of frontier models to contribute decisively to this sector of technological advancement.

3. The core challenges of ARPES (finding patterns in noisy, multi–dimensional datasets) recur across many subfields in physics, and is thus **a good proxy for its other experimental techniques, particularly where spectra and images are analyzed**. ARPES datasets are large enough to challenge context window limits of current models, while not being too high–dimensional such that rudimentary tasks would exceed LLM limits, as in calorimetric data in high–energy physics (Baldi et al., 2016).

In SciPro Arena, models extract patterns by learning from examples rather than apply explicitly stated rules, and return numerical predictions scored by their deviation from the ground truth. High–resolution datasets that push context length limits were generated by a high–fidelity ARPES spectrum simulator to avoid training contamination. We test recent reasoning models with direct query and find that only frontier systems released after December 2024 show meaningful progress. While there is a trend of newer models performing better, even the best models achieve only 0.13 out of 1 averaged over all spectra (Fig. 1, red bars), rising to 0.20 when questions with noise were excluded (blue bars). Enabling code generation increased scores to as much as 0.23 (all spectra) and 0.39 (noiseless), with strong gains for relatively noise–free questions but minimal improvement with severe noise (Fig.1, inset). Within the latest slew of open–weight models, Qwen3 was by far the best, scoring 0.09 (averaged) and 0.18 (noiseless). In comparison, 400 lines of basic code written by a graduate student over three days (Appendix E and supplementary code) already achieve 0.37 (all spectra) and 0.55 (noiseless) — a significant gap in performance. Three capability tiers are revealed: models can extract simple features but fail at tracing continuous patterns or computing derived quantities, the latter constituting core reasoning skills needed for real scientific analysis.

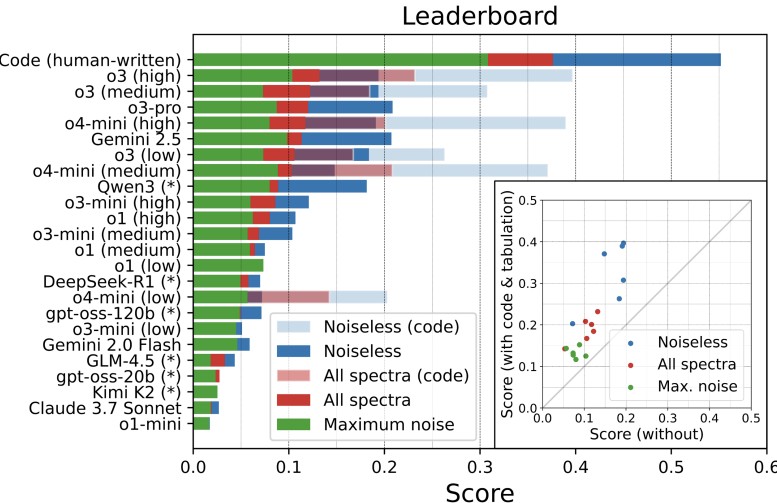

Figure 1: More recently released, closed–weight (proprietary) models perform better. Models with asterisks (*) are open–weight. Note expanded horizontal axis scaling of the score, which tops out at 1. Inset: enabling access to an external Python interpreter and uploading datasets as .csv files improved scores, particularly for questions without noise, but gains are marginal with severe noise.

## 2 RELATED WORK

Scientific benchmarks acknowledge that the scientific process is an iterative cycle of interrelated tasks—hypothesis generation, experiment execution, data analysis, and communication—and correspondingly test either specific tasks in that cycle, or the whole process itself. Regular scientific use of LLMs has now shifted beyond mere brainstorming/writing aids (Liang et al., 2025), and recent benchmarks have sprung up measuring the competence of models in some of these tasks, particularly by crystallizing them as code generation (Chen et al., 2024) and workflow derivation problems (Majumder et al., 2024). Efforts have been made towards automating the full research pipeline (Jansen et al., 2025; Lu et al., 2024; Yamada et al., 2025). The question of whether data analysis can be tackled had been treated by Liu et al. (2024b) on smaller, non–scientific datasets.

SciPro Arena is targeted at the weakest link in the scientific process: it asks whether models can be *trusted* on data analysis, emphasizing maintenance of full standards of rigor, accountability, and interpretability. It is crucial to note that the role of an agent or human in scientific discovery lies in the act of 'controlled rebellion' (Polanyi, 1962), the *originality* or degree of surprise of their discovery held in tension against high standards of *proof* enforced by the inherently conservative attitude of science towards modifying consensus — standards judged and upheld by other human scientists. Any agent seeking to supplement, let alone supplant, the work of a human in science must *first* match the same high bar, or find its results disregarded by the scientific community.

We concern ourselves with the scientifically and technologically crucial field of condensed matter, in particular the most powerful tool in its arsenal — ARPES. Models performing well on SciPro Arena can automate substantial work for this field, and results may generalize to other spectral data analysis tasks in physics and chemistry. Although there is a long history of benchmark development (summarized in Appendix A), we draw from several specific prior works: inductive reasoning in InductionBench (Hua et al., 2025) and (indirectly) in ARC–AGI (Chollet et al., 2025), information extraction from complex data (visual reasoning benchmarks), noise robustness in NoiseQA (Ravichander et al., 2021), and long context processing in Long Range Arena (Tay et al., 2020). We build most directly on Michelangelo's Latent Structure Query framework (Vodrahalli et al., 2024), noting the close parallels between scientific analysis and many standard AI tasks: both involve measuring one thing ('$x$') to learn about something else ('$y$'). Video models, for example, measure pixels ('$x$') in order to learn about objects in the world ('$y$'). Due to the scarcity of real–world quantities *in science* that can be measured directly (experimental data '$x$'), most scientific experiments necessarily involve a proxy relationship between (or representation of) $y$ by $x$, which is inevitably complicated by experimental artefacts. The scientific analysis process, where relevant information ('$y$', represented in the 'domain language' of a field) is hidden rather than obvious, closely resembles Michelangelo's latent structure tasks, which go beyond plain 'needle in a haystack' retrieval of information.

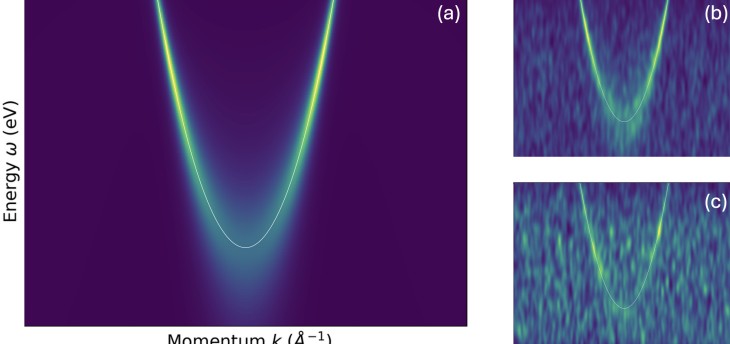

Figure 2: Sample spectral function $A(k, \omega)$ from the bottom of a band, for which three different tiers of questions could be posed. Its dispersion, $\epsilon(k)$ (traced out with a solid white line) has a finite broadness set by the linewidth, $\Sigma''(\omega)$. Three noise levels are shown: (a) 0%, (b) 10%, (c) 40%.

## 3 SciPro Arena Benchmark

### 3.1 ARPES and its complications

Our benchmark tests whether models can extract quantitative information from scientific datasets with realistic experimental problems: the presence of noise and convolution. We focus on Angle–Resolved Photoemission Spectroscopy (ARPES), a condensed matter physics technique that measures electronic structure in materials. Each dataset is a 2D intensity map with real energy ($\omega$) and momentum ($k$) axes. Figure 2(a) shows an example. Pixels represent electron count rates across energy and momenta (the spectrum), which reveals electronic behavior in the material: brighter regions show where electrons are more likely to be found. Physicists extract dispersion curves $\epsilon(k) \in \mathbb{R}$ (how electron energy varies with momentum) and linewidths $\Sigma''(\omega) \in \mathbb{R}$ (how broad spectral features are at different energies). ARPES analysis is then an **inverse problem**. We know the forward process — given $\epsilon(k)$ and $\Sigma''(\omega)$, we can compute the spectrum,

$$A(k, w) = \frac{\Sigma''}{(\omega - \epsilon)^2 + (\Sigma'')^2}.$$

In the language of section 2, this equation maps $y \mapsto x$, where the spectrum $A(k, \omega)$ is the quantity measured ('$x$'), while dispersion $\epsilon$ and linewidth $\Sigma''$ are the ground–truth quantities we wish to extract ('$y$'). The aim of ARPES data analysis is then to work out $x \mapsto y$ ('given a noisy spectrum, extract the underlying dispersion and linewidth functions'), which is harder. Realized by an accurate spectrum simulator written expressly for this benchmark, ARPES makes a good benchmark because:

1. A clear, built–in ground truth exists,

2. Realistic experimental noise can be controlled,

3. Difficulty is adjustable, and

4. We can measure not just whether answers are right or wrong, but how far off they are from the ground truth.

The key attributes of ARPES data analysis complicating the solution of this inverse problem (that is, noise and convolution) recur for many modalities of data defined over continuous domains, the form that predominates in physics and chemical spectroscopies. In astronomical images, one encounters corruption by Poisson noise (Shamshad et al., 2018) convolved with point spread functions, such as the characteristic speckles of the James Webb Space Telescope (Kinakh et al., 2024). The study of jets in high–energy physics likewise involves problems of sifting through background noise (Sjölin, 2012) and accounting for calorimetric instrumental resolution (Lobban et al., 2002).

### 3.2 Structure of dataset and questions

We pose 27 question types, each tested at five noise levels, for a total of 135 questions. Each question uses few–shot prompting (often 3–shot) where models read example spectra with answers to figure out an analysis method, then analyze a spectrum — similarly to ARC–AGI (Chollet et al., 2025).

**Form of questions** All questions were stated in the form of text strings, beginning with the prompt itself and followed by datasets corresponding to example and test spectra, as in this example:

---

Four datasets showing ARPES spectra are contained. They are labeled "Dataset A", "Dataset B", "Dataset C", and "Dataset D". Read "Dataset A". The Fermi energy of "Dataset A" is 2.71 eV. Read "Dataset B". The Fermi energy of "Dataset B" is 15.98 eV. Read "Dataset C". The Fermi energy of "Dataset C" is 8.01 eV. Now read "Dataset D". State the Fermi energy of "Dataset D" in units of electron–Volts. Print only your numerical answer.

Dataset A

Energy (eV) / Momentum:  $-1$  $-0.973$  $-0.9459$  $-0.9189$  $-0.8919$  ...

2.5   513   622   561   609   633   753   619   685   513   566   727   520   493   548   ...

2.504   566   644   600   654   671   818   667   760   544   557   811   560   514   ...

---

All (example and test) datasets within a question are contained in a single text string. The first row of each dataset (after 'Energy (eV) / Momentum:') states the momentum corresponding to each column, while the first (leftmost) column states the energy of each row. The remaining entries list spectral intensities per pixel for the range of momenta and energies contained therein. To reduce token count while still retaining a high dynamic range, spectral intensities are normalized such that the highest intensity is exactly 1000, and all intensities are rounded to the nearest integer.

### 3.3   Types of question

**Quantity extracted**   The 27 categories of questions are grouped under five scientific domains. Because this classification is primarily relevant to condensed matter physicists, we leave fuller explanations in Appendix E, particularly the captions of Figs. 10–14.

**Data analysis tasks**   Each category involves *at least* one of four analytic tasks: **regression** (when simple mathematical formulae can be fit), **structure determination** (objects in spectra which are not straightforwardly mathematically described), **noise dependence** (all questions), and **categorization** of objects. (Fuller explanations of these tasks are given in Appendix E.) Some tasks common in physics are not covered, such as anomaly detection and time series prediction.

**Difficulty tiers**   In parallel, we classify questions into three tiers of difficulty: **Tier I.** Extraction of a single quantity departing to a limited extent from 'needle in a haystack'–type questions. **Tier II.** Extraction of an array of quantities, such as dispersions $\epsilon(k)$ and linewidths $\Sigma''(\omega)$. **Tier III.** Single quantities indirectly determined (calculated after extracting such arrays as those of Tier II). The spectrum shown in Fig. 2 is illustrative. It may be used as the basis to ask questions from three different tiers of difficulty. Asking for the band bottom energy (bottom of parabola) constitutes the easiest, Tier I. Tracing the dispersion (white line) would be Tier II. Asking for the Fermi velocity $v_{\mathrm{F}}$, which is the gradient at the top edge of the parabola, is the most difficult, Tier III, answered rigorously only after having traced the dispersion (Tier II). Note that for questions requiring array–type responses, analytic forms of the ground truth are 'smooth' and remain unchanged no matter how much noise is added. For few–shot prompting, the 'smooth,' noiseless ground truth for example spectra are given to the model to guide its deductive reasoning. These tiers scale differently with token count (or as a proxy, number of pixels = resolution *squared* per spectrum). The complexity of questions in Tier I remains approximately constant with token count, excepting complications from noise, while those of Tier II scale linearly with resolution, therefore approximately as the *square root* of token count. If rigorous analysis is carried out by an agent, questions in Tier III should scale *at least* at the rate of Tier II, with whatever additional scaling is associated with the computation required to extract key quantities from a 1D array.

### 3.4   Real–World Complications: Noise and Convolution

Noise was inserted in the manner of Fig. 6(a) in Kim et al. (2021): as a set of $2 \times 10^5$ randomly–distributed spots, whose intensities are randomly chosen within a range, and footprints broadened into 2D Gaussians. The amount of noise is quantified as *mean* noise intensity as a fraction of *maximum* spectral intensity prior to adding noise. We do not measure against the mean spectral intensity, as this quantity varies with the size of the spectral feature investigated relative to the size of the whole dataset, whereas maximum spectral intensity does not; we are interested in the extraction of strong signals regardless of how much other information (the background signal) is present. Representative noise levels are shown in Figure 2; for most question categories, noise levels scale up to $40\%$. Additionally, all our data were convolved with 1D Gaussians of width 0.005 /Å in momentum and 0.003 meV in energy. This is present in all real–world experimental data. At such a low level of convolution, only minor artefacts such as slight deviation of observed dispersions from the ground truth are produced.

# 4 EXPERIMENTS

## 4.1 SET–UP

**Few–Shot Prompting**   We had observed, during preliminary tests, that models sometimes regurgitated the answers of given examples rather than reason through to obtain the actual answer of the question itself. We have therefore made sure that the correct answer does not overlap or coincide with those of prior test examples (whether they come in the form of a single number, or an array), so that such regurgitation would not accidentally inflate the score. Few–shot prompting was introduced to reduce dependence on elicitation (in which case performance would depend on the skill of the evaluator rather than the model), an approach following that of ARC–AGI (Chollet et al., 2025). Notwithstanding the higher importance models may place to information at the start of prompts (Liu et al., 2024a), the vast majority of content by token count are comprised of the examples themselves ($\sim 10^5$ tokens) rather than worded starting instructions (a few hundred tokens at most). Additionally, we note that dependence on the number of examples could not be independently investigated. Because of large token count, varying the number of examples would introduce trade–offs: either reducing resolution, or performance scaling with token count (see Effect of resolution, Section 4.2).

**Evaluation**   A clear ground truth exists for each question, whether in the form of a single number, several, or an array of numbers consisting of outputs of some analytic function over points in a domain (energy $\omega$ or momentum $k$). Models are prompted to respond by returning a number or set of numbers. There is no numerical restriction to their answer other than the expected array length (which is clearly stated at the start of each question; an array of incorrect length returned is counted as an incorrect answer). As a result, responses are not scored as strictly correct or incorrect. We score each response using a rescaled Lorentzian as a measure of deviation from the ground truth. The score is averaged amongst multiple responses to the same question, then averaged amongst all 135 questions. For a single scalar answer, this takes the form (see Appendix D for details)

$$\text{Score} = \frac{\gamma^2}{(x - x_0)^2 + \gamma^2}$$

for model response $x$, ground truth $x_0$, and half–width half–maximum (HWHM) $\gamma$, such that a completely correct response is scored 1, and a completely *in*correct response 0. As an interpretative rule of thumb, an answer deviating by a typical experimental error (HWHM $\gamma$) scores 0.5. For array responses, this is repeated for each element of the array, then averaged. Depending on the domain of the expected response, HWHM $\gamma$ takes the convolutionary values of $0.005/\text{Å}$ in momentum, $0.003$ meV in energy, and $0.03$ in doping (a quantity explained in Fig.14 of Appendix E). We recognize that more sophisticated measures of deviation (or even distance) may be in principle be used (Appendix D), and that there is no *a priori* reason why a Lorentzian should be selected over, say, a Gaussian; its 'long tail' merely allows us to pick up responses deviating further from the ground truth.

**Models and Inference–Time Compute**   Preliminary tests had indicated that non–reasoning models (prior to December 2024) by and large performed poorly on our test. We have thus restricted our leaderboard to reasoning models released December 2024 and after. Firstly, this includes various closed–weight models from OpenAI (o1-mini, o1, o3-mini, o3, o3-pro, and o4-mini), Google (Gemini 2.0 Flash, Gemini 2.5 Pro Preview), and Anthropic (Claude 3.7 Sonnet). OpenAI models with adjustable inference–time compute (all excluding o1-mini and o3-pro) were evaluated as a function of compute mode (low, medium, and high). Secondly, open–weight models were tested: DeepSeek-R1, Qwen3 (on Thinking Mode), Kimi K2, GLM-4.5, got-oss-120b, and gpt-oss-20b.

## 4.2 RESULTS

**Comparing models**   For a fair comparison, we tested models using direct query on the same low resolution datasets, the highest that would still be accepted by models with the smallest context windows. A clear correlation between release date and model performance was observed (Fig. 1). Much of this may be chalked up to progress in the performance of reasoning models in the recent half–year. In particular, o3 on high compute mode and Gemini 2.5 Pro Preview are among the best–performing models; these likely reflect the long inference time and long built–in context window that the respective models afford. Among open–weight models, Qwen3 (on Thinking Mode) performed competitively for noiseless questions without noise, but performance worsened rapidly with noise.

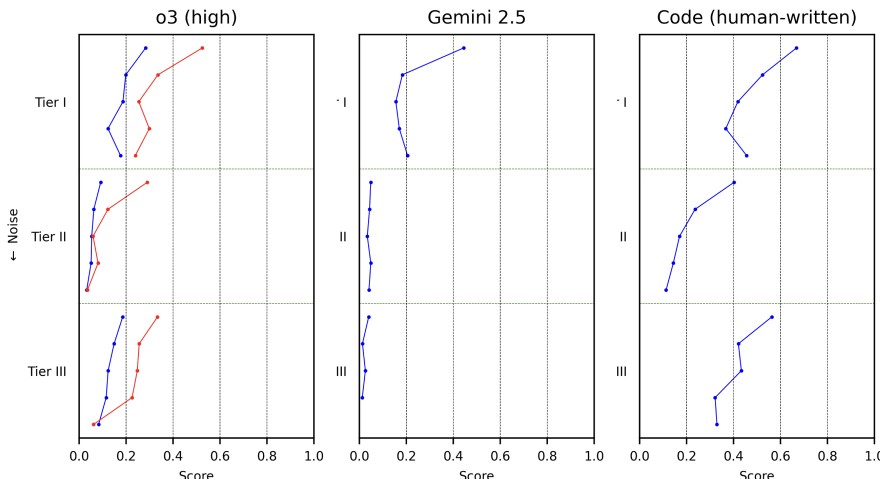

Figure 3: Higher noise worsens performance across tiers. Red: access to a Python interpreter and .csv tabulation improved the scores of o3 (high), but gains were minimal at high noise intensities.

**Comparing tasks**  A similar ordering of model performance was observed for the four data analytic tasks mentioned in Section 3.3, and presented in Figs. 7 and 8 in Appendix F, although a few surprises have showed up: Gemini 2.5 performed surprisingly poorly in tasks involving regression and categorization (relative to its overall performance), and Qwen3 found to be better for structure determination and categorization than other tasks.

**Comparing tiers of questions**  Representative scores from two best–performing models and human–written code are shown in Fig. 3. Full scores are shown in Appendix G. Models performed best in Tier I questions; these did not ask for much beyond an awareness of the context and amounted to a simple retrieval of information, departing not far from older 'needle in a haystack' measures. Poorer performance was observed in Tier II questions, for which a grasp of the underlying latent structure was necessary to answer the question, and in Tier III questions, whose proper analysis would involve two steps models were *not explicitly* guided through: first retrieving a latent structure, then obtaining some information from that structure, such as Fermi velocity or doping level. While Tier III questions can in principle only be rigorously tackled after performing an analysis of a Tier II type, and thus should be at least as difficult as Tier II questions, models may 'short–circuit' the inductive process. This reason, and the fact that our Tier III questions required single–valued rather than array responses, may be why o3 (high) performed similarly on Tiers II and III (Fig. 3, left panel).

**Effect of resolution**  Tests were carried out on how increasing token count (dataset resolution) affected scores in Gemini 2.5 Pro Preview, which had the largest context window. These were limited to two noiseless questions, A1 (a Tier I question) and B1 (a Tier II question), and to the best–performing model for noiseless questions, Gemini 2.5 Pro Preview. Scores worsened more quickly with increasing resolution for B1 than A1 (Fig. 4(a)), although part of this may be accounted for by the square–root scaling of task complexity with token count (number of pixels) in B1. Additionally, while the score for A1 tapered off towards a finite value (around 0.2–0.3) in the limit of long context window/large resolution, that for B1 appeared to drop precipitously towards zero, indistinguishable from near–random responses under our scoring mechanism. It may be the case that Gemini 2.5 Pro Preview's longer built–in context window offset the longer inference–time compute afforded by its main rival, o3, resulting in their similar placements on the leaderboard (Fig. 1). We postulate that at low resolutions, Gemini 2.5 Pro Preview may be placed at an earlier stage of its resolution–dependent reduction in score compared to o3, consequently suffering less decline in performance attributable to long token count, but further tests are needed to justify this more comprehensively.

**Effect of noise**  Limited work has been done on this front, using four questions tested on Gemini 2.5 Pro Preview (Figs. 4(b)–6), a model which performed best for noiseless questions yet showed a steeper decline in score with increasing noise compared to o3. For Tier I questions, such as A1 and

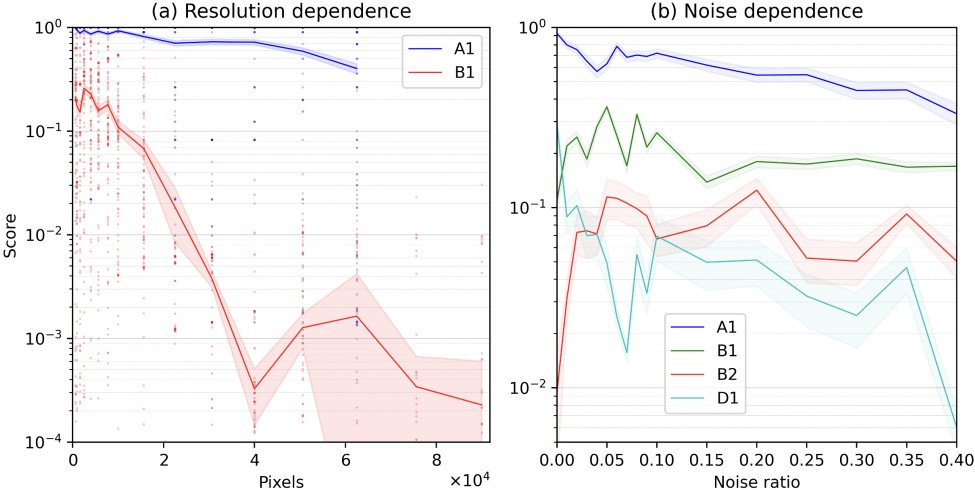

Figure 4: Both (a) higher resolution and (b) stronger noise degrade performance, although a slight gain in performance with modest noise is sometimes present. *Note logarithmic vertical axes (score).* Score distributions are decidedly non–Gaussian, and are instead multi–modal and peaked at certain values, reflecting the persistence of specific responses across resolutions. Shaded areas indicate error in the mean score, also known as the standard error, $\sigma_{\overline{x}} = \sigma_x / \sqrt{n}$ for score $x$, sample standard deviation $\sigma_x$, and sample size $n$.

D1 (at coupling strength $\lambda = 1$), we observed a general trend of scores worsening under increasing noise level, even though this was partly obscured by randomness stemming from the limited number of responses sampled. Scores for D1 were worse than A1 because the feature tested (phononic kink, Fig. 13) was less discernible, particularly at high noise levels. For Tier II questions (B1 and B2), the effect of noise was mixed. The observation that increasing noise results in an 'injection of randomness' in responses generally and visually holds, *especially at higher noise levels*, even if the concomitant decrease in score was not captured by the score due to our strict error criterion (small $\gamma$). In Fig. 5 (a) to (b), and Fig. 6 (b) to (c)), increasing noise clearly led to more responses resembling 'random walks.' It was sometimes observed that *at low noise levels*, a small injection of noise appeared to 'kick' the model out of an incorrect answer and unexpectedly improved the score. This was the case for B2 in Fig. 6, where a completely incorrect response for low noise in (a) (its quadratic dispersion hardly traced) resolved to a less–incorrect response in (b) (the dispersion now approximately traced), before the aforementioned 'random walk' set in at higher noise levels in (c) and severely degraded its responses.

**Tokenization dependence** Errors in numerical calculations are known to stem from how numbers are represented (Levy & Geva, 2025). To isolate the effect of tokenization, tests were run varying maximum spectral intensity as a ratio of original normalization (1000), finding no discernible trend in scores for questions A1 and B1 within the bounds of sampling error (Fig. 9 in Appendix F). It is probable that the difficulty of questions are dependent not so much on low–level numerical calculations (on the fine scale of tokenization) as on how models deal with the mix of tasks involved (section 3.3), such that tokenization errors may be considered as another injection of 'noise'.

**Python interpreter and tabulation** Access to a Python interpreter was additionally given to o3 and o4–mini on all three compute modes. Tabular data was presented in .csv and models were prompted to produce code which was in turn input to the interpreter, whose numerical output were scored (Figs. 1 inset, 24, and 25). Gains in score were substantial at low noise, but marginal with severe noise. This may reflect the corresponding increase in difficulty in mapping inverse problems onto code generation problems. At zero noise, an *analytic* solution may exist and is conceivably retrievable by simple code. However, the intrinsically ill–posed nature of inverse problems rears its head with higher noise, requiring increasingly sophisticated solutions to tame its difficulties, possibly indistillable into a compact coding problem, instead calling for advanced statistical

treatment (Benning & Burger, 2018) or dedicated model training (Ying, 2022; Peng et al., 2020).

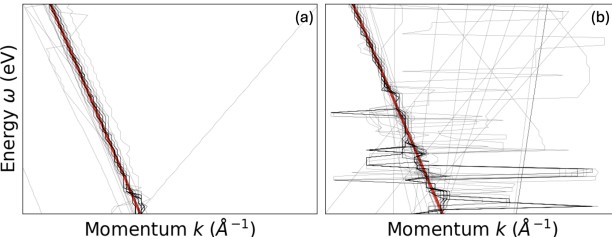

Figure 5: Response accuracy generally declines with increasing noise. A solid red line indicates the ground truth. Strength of noise set at (a) $4\%$ and (b) $35\%$ of maximum dispersion intensity, for a linear dispersion (`B1`, green in Fig. 4(b)).

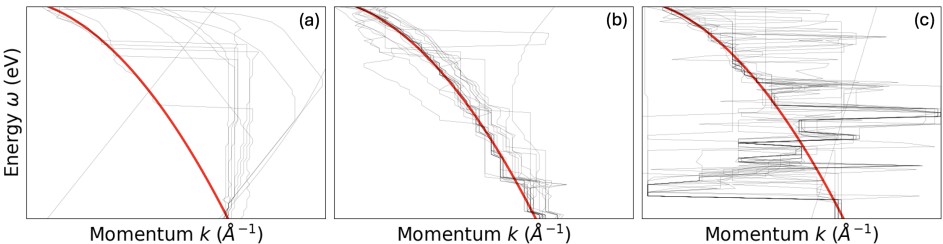

Figure 6: Unexpected improvement and subsequent expected decline in accuracy of answers with increasing noise; ground truth in red. Noise set at (a) $0\%$, (b) $10\%$, and (c) $40\%$ of maximum dispersion intensity, for a quadratic dispersion (`B2`, red in Fig. 4(b)).

## 5 DISCUSSION

We have presented SciPro (Scientific Process) Arena, a benchmark for testing frontier AI systems on analysis of spectral scientific data. We test models on regression, structure determination, noise dependence, and categorization in datasets — core skills for scientific reasoning in physics. Our results show that current frontier models (as of September 2025) can handle simple retrieval tasks but fail at complex pattern recognition and derived calculations that real scientific analysis requires. The best current models score only 0.13 out of 1 on average under direct prompting, and 0.23 when given a Python interpreter and .csv–tabulated data, with clear performance degradation as data resolution increases and noise levels rise. Models perform reasonably on Tier I questions (simple information retrieval) but poorly on Tier II (latent structure extraction) and on Tier III (derived calculations from extracted patterns). This reveals fundamental limitations in how current models process structured numerical information and perform multi–step reasoning on scientific data: the difference between Tier II and Tier III questions arises not from a leap in computational complexity, but rather from combining separate tasks, and unprompted pivoting from the first (not explicitly stated) task to the second, with successful completion of the former required to succeed on the latter.

It is clear that current frontier agents are not ready for publication–level data analysis, where accuracy is critical. However, they could be useful for preliminary analysis, especially when processing large datasets at speed is important. A key limitation in scientific progress is data quality itself; sophisticated analysis cannot compensate for poor data. Reinforcement learning–based agents working in real–time could therefore be valuable, by analyzing large datasets quickly and suggesting experimental adjustments while data is being collected, rather than during post–processing. In the immediate future, this points toward benchmarks that test scientific agents in dynamic, real–world experimental scenarios, akin to $\tau$–bench (Yao et al., 2024), rather than static data analysis.

We anticipate that several developments are necessary before the full potential of agents is harnessed in condensed matter. Firstly, agents should become fluent in the 'domain language' of any field to

be able to represent a system in its full complexity and be poised to make profound inferences, compared to agents with a merely superficial grasp. Having found that code generation by agents is still insufficient, we anticipate fine–tuning models through RL as a next step, or giving agents access to train other (possibly simpler) models (Ying, 2022). In the future, we may need to construct domain–specific foundation models, or even find that transformer–based LLMs themselves are insufficient, calling for new architectures such as world models (LeCun, 2022).

Secondly, agents must be able to reason through different layers of representations. Due to the ubiquity of **emergence** in condensed matter (Appendix C), the microscopic origin at which theories work is typically *many (qualitatively) different layers of reasoning removed* from measurable properties, the latter of which are the starting point of data analysis. The path from experiment to theory is therefore highly serpentine and at times impenetrable. (ARPES is special in that the number of these 'layers' involved is fewer than most other experimental techniques.) The scientific process, especially in condensed matter, consists not of a *single* inverse problem (Tier I and II questions in SciPro Arena), but of a *string* of inverse problems, in which the description of the system is couched in a different representation along each step. The ability to start from raw data and reason through to the level of microscopic theory, may be a necessary precondition for a model to tackle the problem of emergence. Until that point is reached, human intervention is expected at every step to guide models through these disparate layers of reasoning.

Lastly, an agent has to be capable of generalizing and reasoning *across* the results of various measurements to piece together a physical picture that transcends the mere specifics of individual experiments — today, this is achieved through the formation of a scientific consensus across a broad array of human experts (Polanyi, 1962). This is because an *irreversible loss of information* occurs in data collection, as the complexities of microscopic physics are collapsed into the highly constrained feature space that each measurement affords. The parable of the blind men and the elephant is apt. From a *single* experiment alone, one cannot uniquely trace any phenomenon back to a single underlying microscopic origin, therefore a holistic description of a system or phenomenon (such as a superconductor) is necessarily pieced together from *disparate* experiments; even this collected information provides but a fragmentary understanding. We hope that agents in the medium–term future, possibly acting in synergy with human guidance, may supplement this process by performing a variety of targeted measurements to compare with, and constrain possible physical models. In superconductivity, for example, structural probes (such as X–ray diffraction) and electronic probes (such as ARPES) may be compared with theoretical predictions arising from microscopic mechanisms (phonons, spin fluctuations, excitonic pairing). To maintain rigor in the scientific process, and achieve convergence between theory and experiment at each step, critical for preserving a coherent overall understanding, each of these quantities must be benchmarked and calibrated with experiments.

We wish to extend our benchmark to encompass data from other experimental techniques in condensed matter physics as well as elsewhere in physics, and welcome future scientific collaborators. Experts in experimental techniques who may implement these extensions may adopt approaches which are likely to be different from us. Therefore, we have formatted questions in SciPro Arena in a *general yet standardized manner* to ensure consistency of application and analysis, while allowing freedom in realizing details (details in Appendix B).

## REPRODUCIBILITY STATEMENT

An anonymized repository containing downloadable source code and collected data is linked here. The user should set up their own API keys, server and database (see instructions in `readme.md`). Bare source code, without data, is also available in the supplementary material.

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

## APPENDIX A    PREVIOUS BENCHMARKS

Developing benchmarks to test the manifold capabilities of AI has been an area of serious inquiry since its inception (Turing, 1950). Early benchmarks drew upon specially collected, publicly available datasets, and tackled specific tasks: MNIST (Deng, 2012) for visual character recognition, ImageNet (Deng et al., 2009) for object classification, SQuAD (Rajpurkar et al., 2016) for reading comprehension, and BLEU (Papineni et al., 2002) for machine translation. A trend towards increasing comprehensiveness is discerned, notably in GLUE (Wang et al., 2018) and SuperGLUE (Wang et al., 2019) benchmarks for natural language processing (NLP), as well as breadth, such as in the UCI Machine Learning Repository (Kelly et al., 2023), MMLU (Hendrycks et al., 2021), and GSM8K (Cobbe et al., 2021).

The recent development of LLMs has spawned increasingly sophisticated benchmarks. Chatbot Arena (Chiang et al., 2024) pioneered the crowd–sourcing of chatbot evaluation to the public, which was extended in SciArena (Zhao et al., 2025) to scientific literature tasks evaluated by the scientific community. The rigor of benchmarks has also improved considerably, evidenced by the increased attention paid to the flaws of earlier generations of benchmarks. A few examples in NLP suffice: Holistic Evaluation of Language Models (HELM) (Liang et al., 2023) recognized diverse preferences of evaluators in vastly broadening the scope of metrics applied, AlpacaEval (Dubois et al., 2025) tackled the bias of auto–evaluators in favor of longer answers, and LiveBench (White et al., 2025) minimized the effect of test set contamination by continually refreshing its corpus of tasks. The scope of tasks evaluated has also evolved, particularly towards real–world tasks in WildBench (Lin et al., 2024) and professional coding in SWE–bench (Jimenez et al., 2024). In parallel, the level of domain expertise tested has grown markedly, such as GPQA (Rein et al., 2023) and Humanity's Last Exam (Phan & et al., 2025) covering graduate–level tasks across diverse fields and FrontierMath (Glazer et al., 2024) targeting expert–level mathematical problems. The past year has seen the emergence of specialized, GPQA–style benchmarks (that is, in the format of graduate school–level examinations) written by domain experts, including the field of physics (Chung et al., 2025) and condensed matter in particular (Wang et al., 2025).

## APPENDIX B    EXTENSIONS TO OTHER EXPERIMENTAL TECHNIQUES

This appendix presents general guidelines for other experimentalists wishing to evaluate the capabilities of agents on analyzing their own data, in the same manner as SciPro Arena.

**Format of prompts**    In adapting SciPro Arena to various experimental techniques, prompts should contain only a modicum of information necessary for agents to understand the nature of the data they are presented, without being assisted by additional information. Researchers with expertise in different fields are disposed to phrase questions very differently, and it is crucial to ensure that in extending the scope of SciPro Arena, the test remains a pure evaluation of *reasoning* abilities of agents themselves, rather than partly be a test of *elicitation* cabilities by the writers of prompts. The format of questions should follow that set out in Section 3.2. Several guidelines apply:

- Questions should be written in `.txt` and structured in the form `prompt + content`, with all data contained within `content`.
- In lieu of a detailed explanation in the `prompt` on how data should be analyzed, agents should deduce this method through few–shot prompting: multiple examples are given (in `content`) with their answers stated (in `prompt`), before the actual dataset to be analyzed is stated (obviously without the answer).
- The expected form and length of the answer should also be stated in the prompt: whether it is a single numerical value or an array of values; for the latter, the length of the array should also be stated. This is necessary for analysis to be automated, while agents with structured responses are not unfairly advantaged over agents without.
- Data should be either one– or two–dimensional, with axis values, units, and labels stated. Two–dimensional data should presented as a table; the explanation in Section 3.2 suffices.
- Precision of numerical values in the data may be left to the discretion of the evaluator. In our ARPES case (which may generalize to other methods with high–resolution,

two–dimensional data), the context window afforded by an agent may be easily exceeded, and we found it necessary to limit the precision of our data in the manner mentioned in Section 3.2.

**Tiering of questions**   We recognize that different experimental probes involve vastly different issues when it comes to the question of interpretability. ARPES is unusual in that variations in intensity in ARPES data may be straightforwardly ascribed to single–particle excitations. That is because it has a comparatively *large feature space* (energy and several momentum axes) which are collapsed into a smaller set of axes in many other measurements: for example, many other spectroscopies (such as Raman, or Angle–Integrated Photoemission) are not resolved in momentum space, and retain only the energy axis (the 'spectrum', hence 'spectroscopy'). Furthermore, many transport measurements (such as resistivity and heat conductivity) lack even energy resolution; instead, data consists of the variation of some quantity with another (such as temperature).

Nevertheless, the tiering of questions by difficulty (Section 4.2) may be generalized to other measurements. Departing from the set of difficulty tiers stated in Section 3.3, examples are given for other experimental techniques, with prior ARPES examples (Appendices E and G) italicized.

- Tier I — extraction of **a single quantity** departing to a limited extent from 'needle-in-haystack' type questions. Examples: *Fermi level, band bottom energy, Dirac cone energy, superconducting gap size, phonon frequencies*, positions of peaks, minima, and discontinuities in 1D arrays such as resistivity/conductivity, specific heat, magnetic susceptibility, quantum oscillations, and most spectroscopies presenting only energy resolution.

- Tier II — extraction of **an array of quantities**. Examples: *tracing 1D dispersions $\epsilon_k$ and line widths $\Sigma''(\omega)$ from a 2D data array*, dispersions in RIXS data, variation in superconducting gap size along a 1D path in tunneling measurements, edge states in microwave impedance microscopy.

- Tier III — single quantities that are indirectly determined; that is, **quantities calculated after extracting** such arrays as those of Tier II, or parsing arrays directly stated in the data. Examples: *Fermi velocities, doping levels*, scaling laws in transport measurements and what processes they may be ascribed to (sources of scattering), scattering wavevectors obtained by Fourier transforming quasiparticle interference data, matching Raman frequencies obtained from the data with their possible origins (such as specific phonons), and matching RIXS dispersions with possible excitations (such as magnons).

**Experimental complications**   It is recommended that data be simulated, with expected experimental complications inserted. This makes sure that (1) a predetermined ground truth exists (and is not restrained by the potential inaccuracy and slow speed of human data analysis), and that (2) the intensity of complicating factors may be quantified. In particular, we watch out for:

- *Noise levels*. In our benchmark, the intensity of noise is measured as a fraction of the *peak signal* intensity, rather than alternatives such as average intensity of the entire dataset, which depends on such other factors as background level and axis ranges. A similar prescription is recommended. (Our only deviation from this is in Fermi level extraction from a featureless background, for which the background itself constitutes most of the 'signal').

- *Convolution*. It is evident that broader convolution increases the difficulty for a signal to be extracted. While we have not conducted a detailed study on this dependence (as we had for noise), we have found it sufficient to fix convolution at a level expected in an actual experiment, and measure deviation with respect to the half–width half maximum (HWHM) of the convolution (Section 4.1).

## Appendix C   Emergence in condensed matter

The core idea that emergent behavior is central to the description of large systems is a point of commonality between the artificial intelligence and condensed matter communities. This is the belief that knowledge of the properties or laws that govern *smaller components* (such as atoms in physics, or artificial neurons in machine learning) are at best ancillary to, and at worst exceedingly

insufficient for predicting the behavior of *large systems* composed of them (by the same analogy—a material, or a neural network). It stands in (partial) opposition to the reductionist impulse that had historically prevailed over much of physics, and is embodied in AI thinking by such approaches as mechanistic interpretability. We provide several examples of emergence for readers in the AI community.

**Criticality and scaling laws**    As a system approaches a phase transition (such as boiling water), fluctuations become increasingly large (size of bubbles), rendering most microscopic details irrelevant to its behavior. This gives rise to scaling laws that relate thermodynamic quantities and depend on very few properties. Vastly different systems that happen to share these properties (more precisely, dimension and symmetry) then develop the same scaling laws; this is known as **universality** (Stanley, 1971). An example is the onset of ferromagnetism and boiling water (critical opalescence), both of which are classed into the Ising model. The idea that scaling laws emerge in large systems is also central to the AI mythos, epitomized by the work of Kaplan et al. (2020), which considered how performance (loss) scaled with some measure of size (such as parameters, tokens, or compute), and recognized the same phenomenon of universality. A key difference exists between the two communities. Scaling laws in AI often revolve around some aspect of system size, as this approach satisfies the need to improve performance far more than varying architectural details (for which similar laws exist). On the other hand, physicists, whose attention is not focused on any single overarching goal in scaling, consider a wider range of laws.

**Superconductivity**    This is an inherently quantum mechanical phenomenon and an example of a collective instability. When conditions arise in which electrons experience a net attraction rather than repulsion, and pair up (Cooper pairing), a lower-energy ground state is favored. This ground state is the superconducting state; it manifests in ARPES spectra as the opening of an energy gap around the Fermi level (see question `B6` in Appendices E and G). Although the mechanism for Cooper pairing is microscopic, the superconducting state exhibits such macroscopic properties as zero resistance, expulsion of magnetic fields, and macroscopic quantum coherence.

**Non–Fermi liquids**    In a Fermi liquid (such as many metals), interactions between electrons are weak enough that the qualitative behavior of the whole system departs little from a hypothetical system that lacks these interactions (the independent electron approximation), albeit with some numerics altered (renormalization). This is because a one–to–one mapping between states of these two systems (quasiparticles and independent electrons, respectively) is retained; a superficial analogy may be made with the process of tokenization in LLMs. In a *non*–Fermi liquid, however, the mapping breaks down, and the system produces *qualitatively different* behavior. This is partly because the fundamental constituents of the system have ceased to be quasiparticles; what they are is presently unknown. In a sense this is a problem of *representation*. Those in the AI community grappling with non–interpretable features in large models are dealing with a comparable matter.

## APPENDIX D    SCORING SYSTEM

It was stated in Section 4.1 that a Lorentzian (otherwise known as Cauchy) distribution was used,

$$\text{Score} = \gamma^2 \cdot \left[ (x - x_0)^2 + \gamma^2 \right]^{-1}$$

with choices of the half–width half–maximum $\gamma$ depending on the quantity being measured. There is no mathematically rigorous reason why the Lorentzian distribution should be favored over others, but two general principles guided our choice of measure:

- Completely correct answers should receive a score of 1 and completely incorrect answers a score of 0; that is, there is a finite bound of scores, and this bound is normalized to unity. Our intention is to reward answers which come close to the ground truth without unnecessarily penalizing those which stray far from it. An answer that is *rather* wrong and an answer that is *very* wrong would *both* attain similar, near–zero scores. This rules out the use of loss or reward functions which are not bounded, as they discriminate between increasingly incorrect answers in their task to guide models towards optimal solutions. (Inverting

this reason, future extensions of SciPro Arena tailored to performing reinforcement learning on open–weight models may replace bounded scores with unbounded loss functions.)

- The score monotonically decreases with increasing deviation. A different measure abiding by this requirement could have been chosen, such as a Gaussian (whose choice could have been supported by the central limit theorem), but this choice only approximately amounts to a redistribution of weighting between answers of different accuracy, and a similar change could be effected by just changing the value of $\gamma$. A Lorentzian was ultimately chosen for two reasons: (1) it has a longer tail compared to the Gaussian, which is equivalent to more relaxed scoring, and (2) it is the Lorentzian and not Gaussian distribution that naturally appears in spectral line shapes such as those seen in ARPES, where half–width half–maxima $\gamma$ do attain physical meaning. Scientists extending this benchmark to other fields may wish to use the Gaussian if they see fit, especially if its use is found to be more justified within their domain.

## APPENDIX E    QUESTION CATEGORIES AND EXAMPLE SCRIPTS

The 27 categories of questions mentioned in Section 3.3 are grouped into five domains (A–E) according to the physical quantity of interest. These are shown in Figs. 10–14 with accompanying explanations. Note that all spectra are shown at the highest resolution setting (at the upper end of the scale in Fig. 4(a)) and *without* noise. Full scores for all models tested, across all 135 questions (multiplexed by five levels of noise), are shown in Figs. 15–20, with explanatory comments. Note that shaded areas in these figures indicate error in the mean score, also known as the standard error,

$$\sigma_{\overline{x}} = \frac{\sigma_x}{\sqrt{n}}$$

for score $x$, sample standard deviation $\sigma_x$, and sample size $n$ for each question.

These five domains of scientific significance are:

- **A.** Extraction of Fermi level (`A1`).
- **B.** Extraction of dispersions $\epsilon(k)$ or information thereof. Questions requiring array responses include linear (`B1`) and curved/quadratic (`B2`) dispersions; linear dispersions with superstructure (`B3`) and displaying band bottoms (`B4`). Questions requiring single numerical answers are the Fermi velocities ($v_F$) of linear (`B1_vF`), curved (`B2_vF`), and superstructural (`B3_vF`) dispersions, as well as the energies of band bottoms (`B4_bbE`), Dirac cones (`B5`), and superconducting gaps (`B6`).
- **C.** Extraction of linewidths $\Sigma''(\omega)$ as arrays, whose variation with energy is an indication of interaction processes in a material: impurity scattering (`C1`), marginal Fermi liquid/MFL (`C2`), Fermi liquid/FL (`C3`), MFL and a single phonon (`C4`), FL with a single phonon (`C5`).
- **D.** Retrieval of phonon energy, whose presence is revealed by a kink in the dispersion $\epsilon(k)$ at the phonon energy and increase in linewidth $\Sigma''(\omega)$ approaching and passing below the phonon energy. This includes a single phonon (`D1`) at five levels of coupling strength $\lambda$ (showing up as increasing salience of the aforementioned phonon features), as well as two (`D2`) and three (`D3`) phonons at a fixed, intermediate coupling strength.
- **E.** Extraction of doping level for a single–band cuprate (`E1`), two–band cuprate (`E2`), strontium ruthenate (`E3`), and three–band nickelate (`E4`).

Four data analysis tasks are identified. With the exception of noise dependence, scores for each task are calculated by averaging over the scores of questions tagged with each task.

- **Regression.** These apply when simple mathematical formula may in principle be fit onto spectra, which covers all questions *except* `B5`, `B6`, and the **D** of questions where the mathematical form is not apparent.
- **Structure determination.** This covers identifying such objects as superstructure in `B3`, `B3_vF`, band bottoms in `B4_bbE`, the Dirac cone in `B5`, separating the superconducting gap of `B6` from the rest of the bandstructure, as well as phononic kinks in the `D` series and Fermi surfaces in the `E` series of questions.

- **Categorization.** These questions involve distinguishing between different objects in spectra and assign different values to them, such as the frequencies of multiple phononic kinks in `D2` and `D3`, and the doping levels of separate bands in `E2`—4.

- **Noise dependence.** Scores are calculated within each question category by taking the ratio of the score corresponding to maximum noise over that of no noise (with a ceiling of unity), multiplied by the score with maximum noise, and averaged over all categories.

Lastly, the three tiers of difficulty may be mapped.

- **Tier I.** Extraction of a single quantity departing to a limited extent from 'needle in a haystack'–type questions; these are `A1`, `B4_bbE`, `B5`, `B6`, and the `D` series of questions.

- **Tier II.** Extraction of an array of quantities, such as dispersions $\epsilon(k)$ and linewidths $\Sigma''(\omega)$, including `B1`, `B2`, `B3`, `B4`, and the `C` series of questions.

- **Tier III.** Single quantities indirectly determined (calculated after extracting such arrays as those of Tier II). These include `B1_vF`, `B2_vF`, `B3_vF`, and the `D` series of questions.

Sample, human–written code used as an equivalent non-agentic comparison is located towards the end of `client/init.py` in the supplementary code, and is summarized here.

- `A1`. Intensity was summed across all momenta to yield curves that depend only on energy (these are known as energy distribution curves, or EDCs). The EDC was then fit with a Fermi–Dirac function (see caption in Fig. 10) to retrieve the Fermi level, $\mu$.

- `B1`–`B3` and `B5`. Spectra were sliced at each individual energy level to yield momentum distribution curves (MDCs, momentum-dependent counterparts to EDCs). Each MDC was fit with Lorentzian(s), and the process repeated across energies to extract their dispersions $\epsilon(k)$. Similar EDC analysis was used for `B4` and `B6`.

- `C1`–`C5`. The same MDC analysis of the `B` series of questions were used to extract linewidths $\Sigma''(\omega)$, which are related to the half–width half–maxima (HWHM) $\gamma$ of Lorentzians. The main source of error in this analysis is that momentum and energy convolutions artificially increase HWHM; retrieval of actual HWHM through deconvolution or other procedures is an active area of research in the field.

- `D1`–`D3`. Linewidth analysis of the `C` series of questions was first carried out. Phonon frequencies were heuristically defined as those at which linewidth increased most rapidly with energy, within certain energy bounds.

- `E1`–`E4`. Fermi surface maps were sliced at fixed values of $k_x$ and the resulting intensities were fit with Lorentzians whose peaks trace the locus of locus of the Fermi surface, whose area is related to the doping level (see caption in Fig. 14). The main difficulty in this procedure is that an awareness of Fermi surface topology is required to interpret the fitted curves. As a result, human intervention is required in professional practice, and the present (unagentic) code performs relatively poorly.

Note that the standard of the field is significantly higher than the given code, but the present standard of code was chosen for two reasons:

1. Professional data analysis in ARPES requires almost continual human intervention (e.g. judging goodness of fit, removing erroneous data), whereas we prefer that the code be self–contained and transparent.

2. What constitutes the standard of the field also depends on the specific analysis task at hand; being an inverse problem with noise, it remains an active area of research across many diverse fields.

## APPENDIX F   ADDITIONAL TESTS AND LEADERBOARDS

Figures showing the effect of tokenization (Fig. 9) and breakdown of scores by data analytic tasks (Figs. 7 and 8) from Section 4.2 (Results) are shown here.

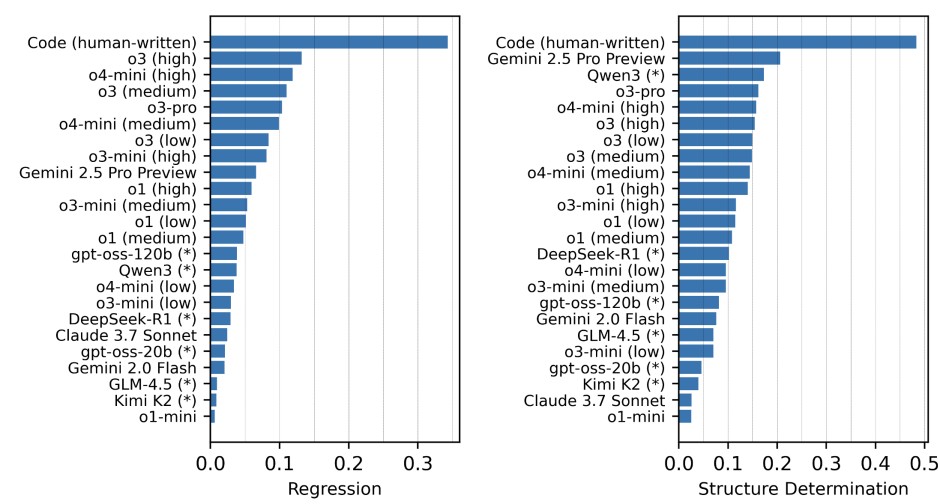

Figure 7: Leaderboards for regression and structure determination.

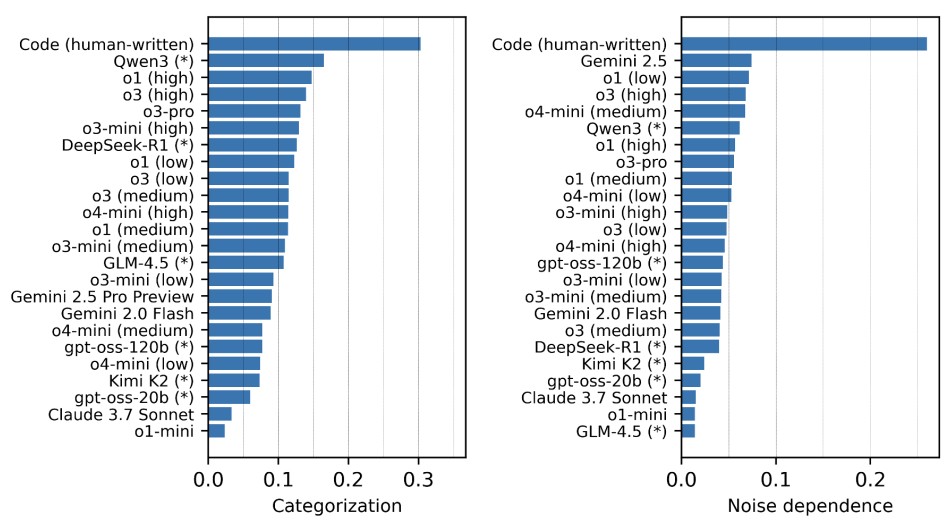

Figure 8: Leaderboards for categorization and noise dependence.

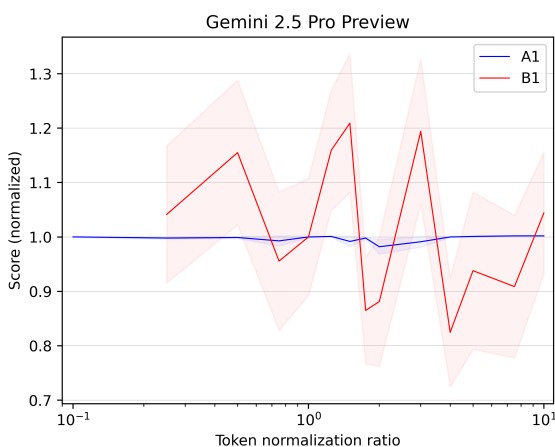

Figure 9: Effect of normalization of maximum spectral intensity on scores for `A1` and `B1`, relative to their score at the original normalization value of 1000. No clear trend could be confidently stated within the bounds of sampling error (shaded areas correspond to one standard deviation in the mean).

## APPENDIX G   QUESTION DOMAINS AND FULL SCORES

The remaining pages of this text are a catalog of noiseless spectra representative of question domains, and breakdowns of scores for all models.

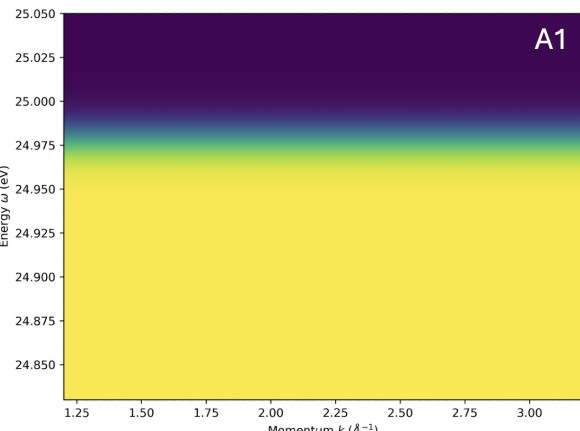

Figure 10: Question domain A (Fermi level extraction). This is the easiest question domain, judging from the full scores in Figs. 15–20. Spectra here (`A1`) are featureless in momentum $k$ (horizontal axis), and take on a Fermi–Dirac distribution in energy $\omega$ (vertical axis),

$$\text{Intensity} \propto \frac{1}{e^{(\omega-\mu)/k_{\mathrm{B}}T}+1}$$

where $\mu$ corresponds to the Fermi level, $k_{\mathrm{B}}$ is the Boltzmann constant, and $T$ temperature. The Fermi level is the midpoint of spectral intensity and is visually obvious in the figure; the point of `A1` is to retrieve this energy as a single number, a task not far removed from a 'needle in a haystack' retrieval, although the addition of noise complicates this somewhat. Note that spectra in question domains B–D (Figs. 11–13) have an upper cut–off at the Fermi level as little information of interest is contained above the Fermi level; this also separates the problems tested by later questions from the task of Fermi level extraction itself. Question domain E (Fig. 14) is arrayed along two momentum axes, cut at the Fermi level in energy $\omega$.

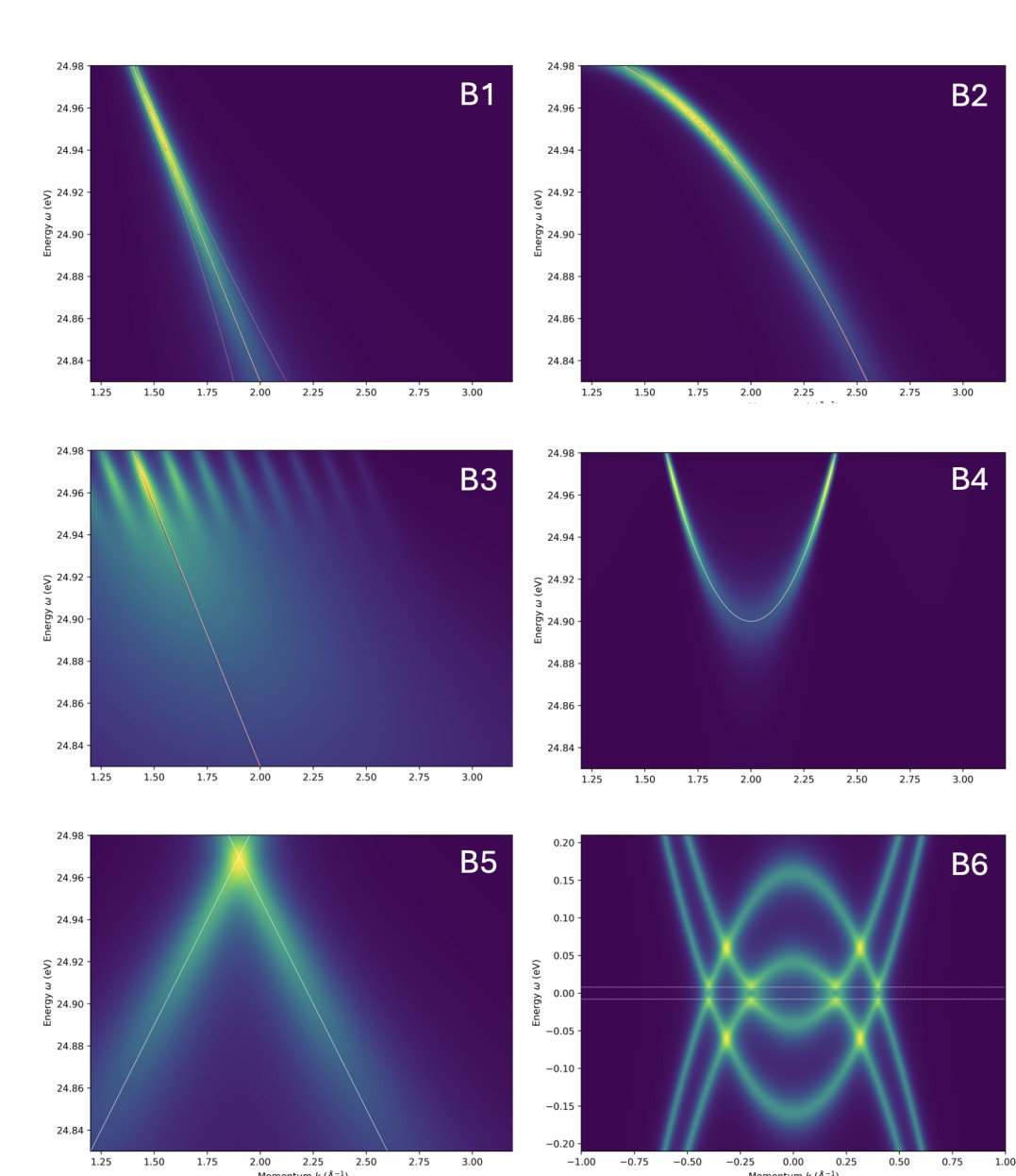

Figure 11: Question domain B (dispersion tracing). These cover a linear dispersion (`B1`), curved/quadratic dispersion (`B1`), linear dispersion with superstructure (`B3`), a quadratic dispersion with a band bottom (`B4`), a Dirac cone (`B5`), and superconducting gap(s) (`B6`). For `B1`–`B4`, models are prompted to trace the dispersion itself (white line) as function of momentum $k$ or energy $\omega$; these are Tier II tasks. `B4_bbE`, `B5`, and `B6` respectively ask for the band bottom energy (bottom of parabola), Dirac cone energy (crossing point of two dispersions), and superconducting gap energy (given by half the distance between the two horizontal white lines); these are Tier I questions. Lastly, `B1_vF`, `B2_vF`, and `B3_vF` ask for Fermi velocity $v_{\mathrm{F}}$, which are the gradients of the dispersion at the top of the spectra; these are Tier III.

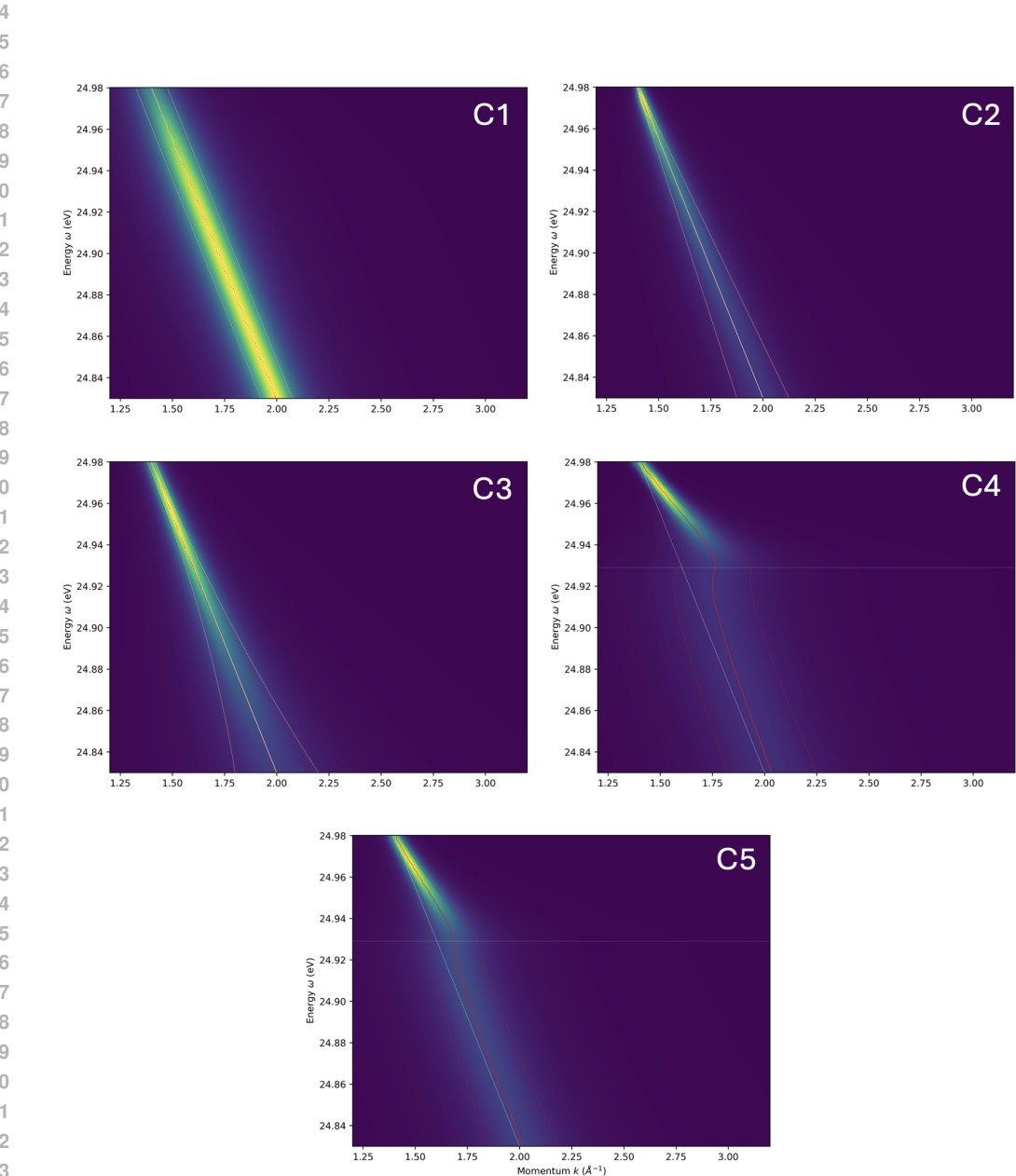

Figure 12: Question domain C (linewidth tracing). Models are prompted to reproduce half–width half–maximum (HWHM) as a function of energy $\omega$, visually shown as half the distance between the red lines. These cover cases where HWHM is constant (C1, impurity scattering), increases linearly away from the Fermi energy/top of spectrum (C2, marginal Fermi liquid), and increases quadratically (C3, Fermi liquid). C4 and C5 are variants of C2 and C3 with the presence of a single phonon at some phonon energy (horizontal white line). The presence of the phonon is signaled by a shift in the (red) dispersion away from the original linear (white) dispersion, an effect termed mass renormalization by physicists, and a broadening of linewidth below the phonon energy.

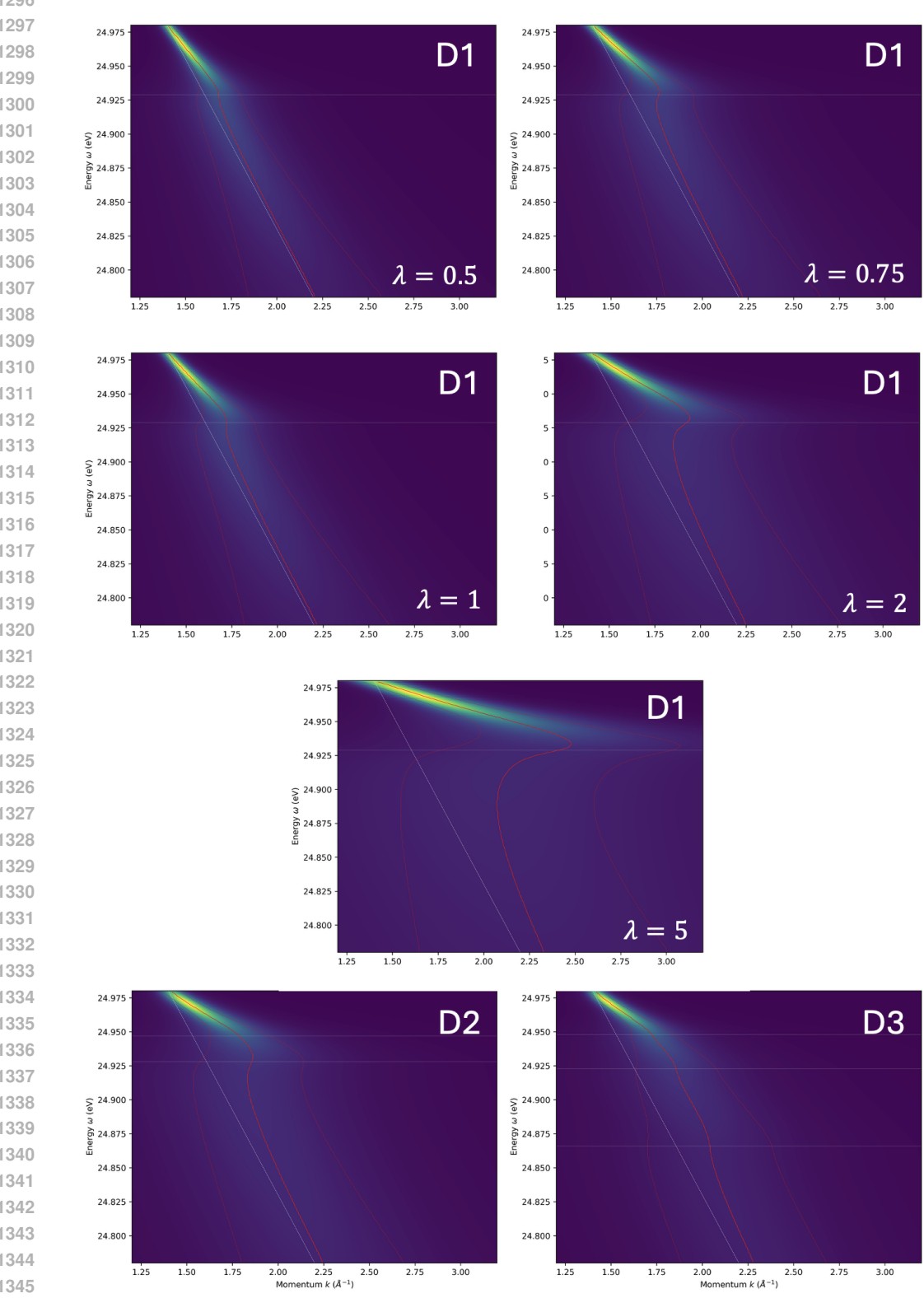

Figure 13: Question domain D (phonon energy determination). Horizontal white lines indicate phonon energies for a single phonon at five different coupling strengths $\lambda$ (D1), as well as two (D2) and three phonons (D3); models are prompted to state these energies.

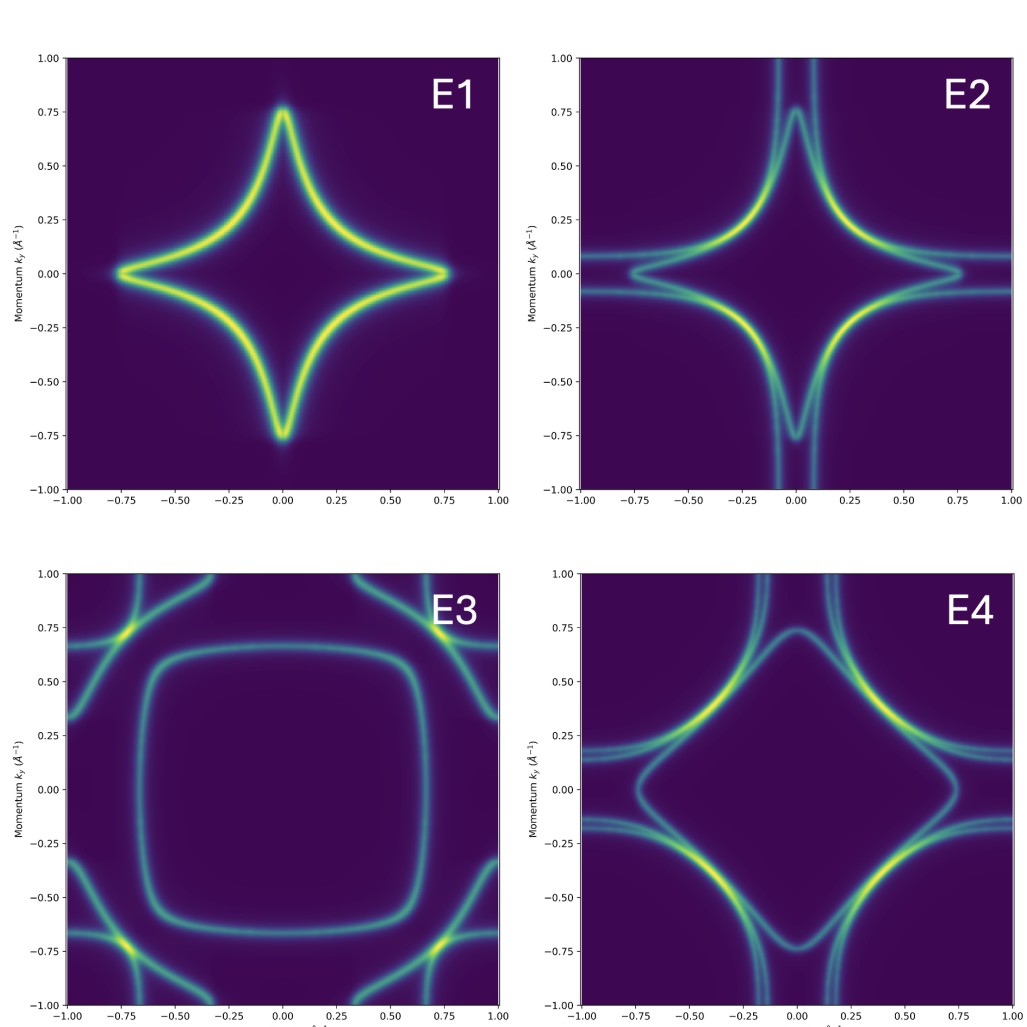

Figure 14: Question domain E (doping determination). Spectra shown are Fermi surface maps in two momentum axes, $k_x$ and $k_y$, over a single Brillouin zone (BZ). Fermi surfaces enclose areas whose size is linearly dependent on the doping level of a material. To keep matters internally consistent, we normalize each BZ to the same extent ($\pm 1/\text{Å}$ in both directions), and define

$$\text{Doping} = 1 - 2 \cdot \frac{\text{Area}}{\text{Area}_{(\text{BZ})}},$$

such that a Fermi surface that takes up the whole BZ has a doping of $-1$ (completely electron–doped), and none of the BZ, $+1$ (completely hole–doped). Suffice it to say that conventions and BZ sizes vary in real life, but we wish to keep things simple here.

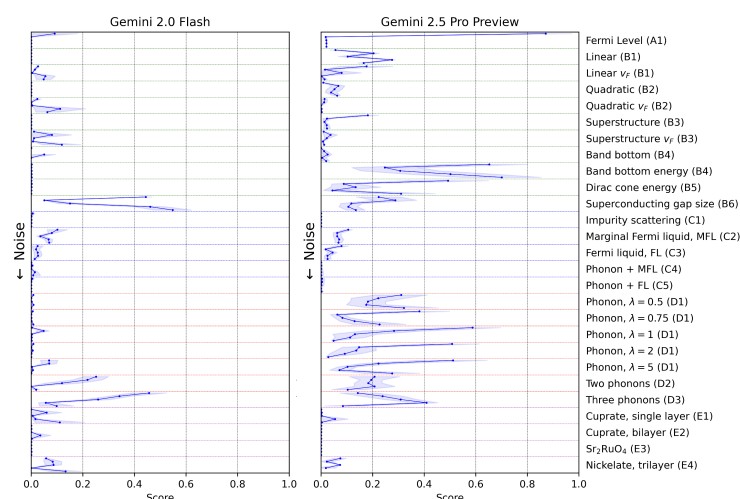

Figure 15: Scores from Gemini 2.0 Flash and Gemini 2.5 Pro Preview.

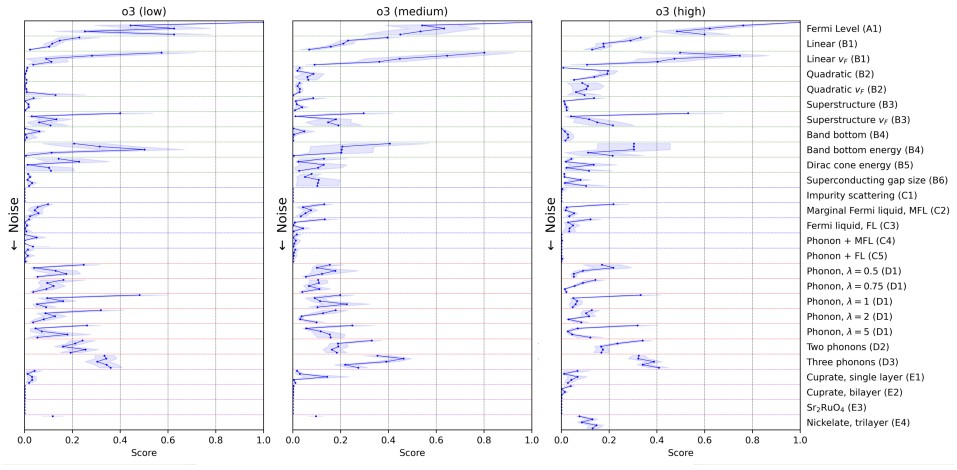

Figure 16: Scores from o3 for three inference–time compute modes.

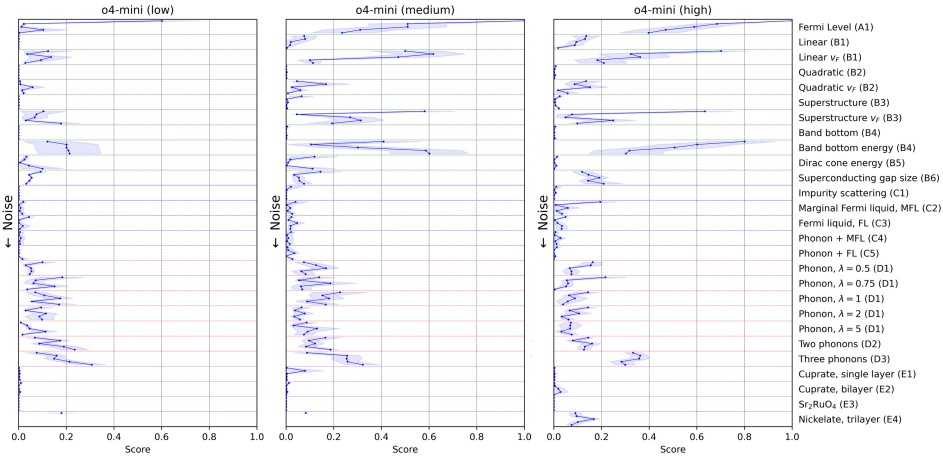

Figure 17: Scores from o4-mini for three inference–time compute modes.

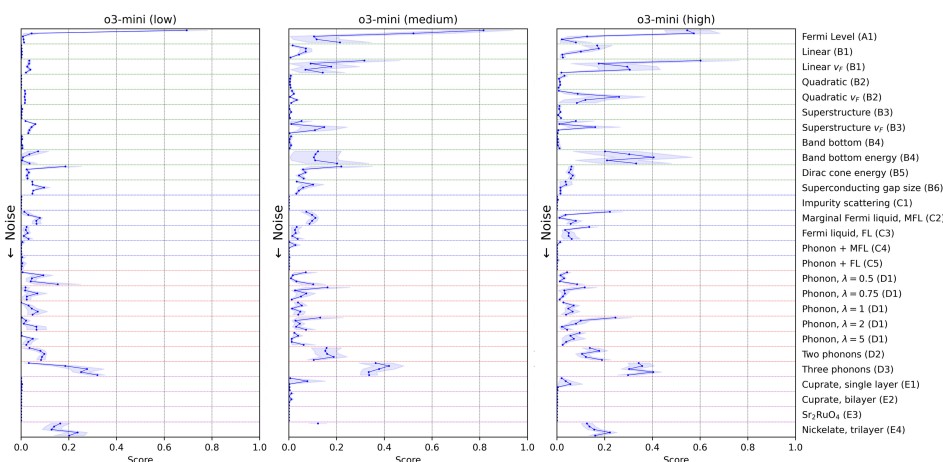

Figure 18: Scores from o3-mini for three inference–time compute modes.

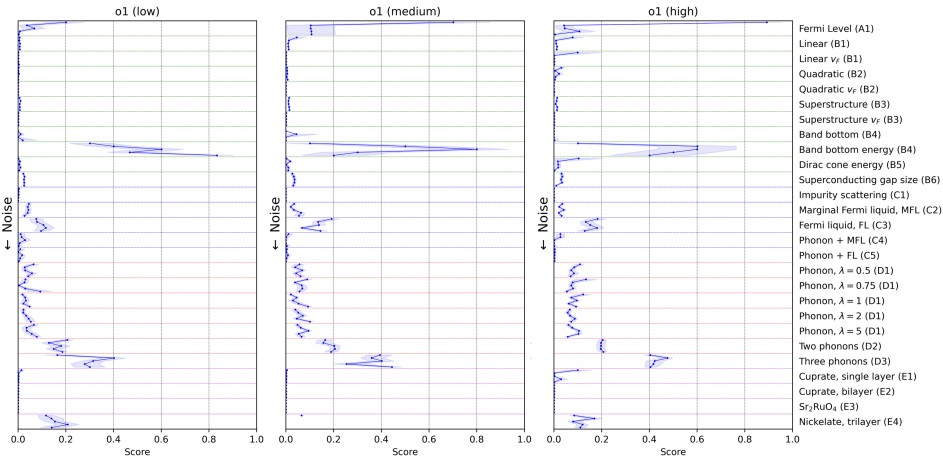

Figure 19: Scores from o1 for three inference–time compute modes.

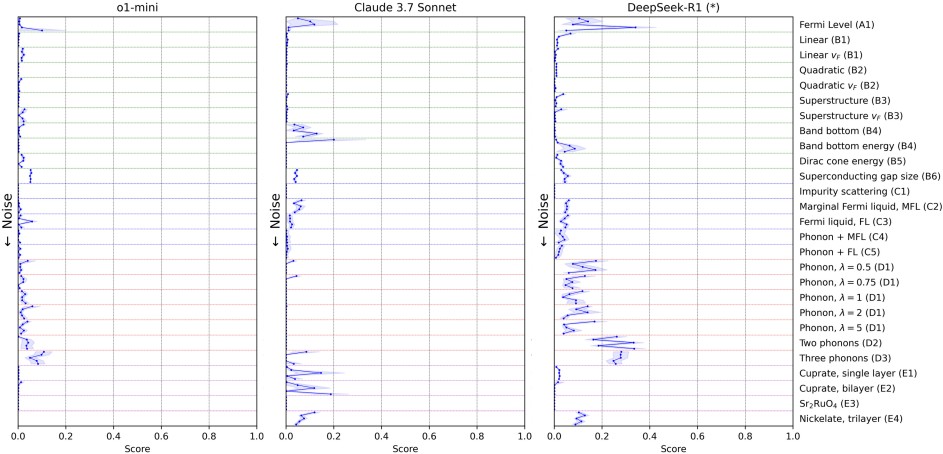

Figure 20: Scores from o1-mini, Claude 3.7 Sonnet, and the open-weight model DeepSeek-R1.

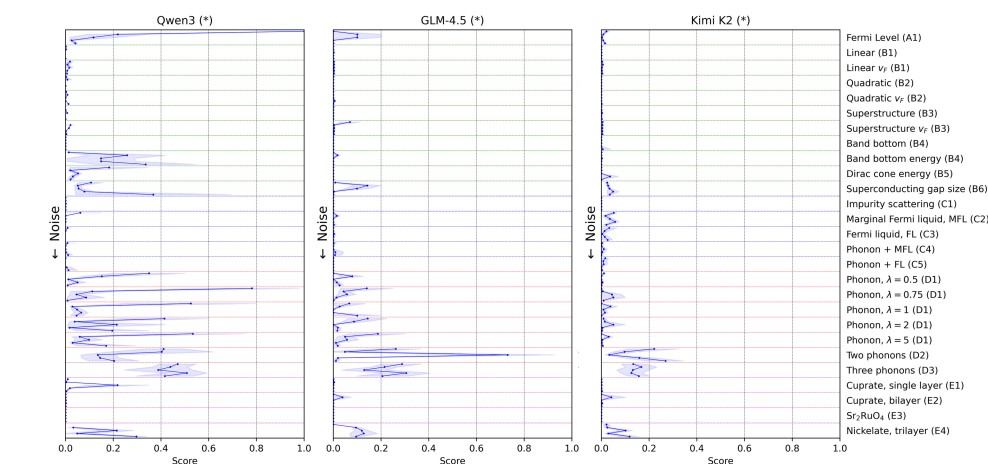

Figure 21: Scores from the open-weight models, Qwen3 (Thinking Mode), GLM-4.5, and Kimi K2.

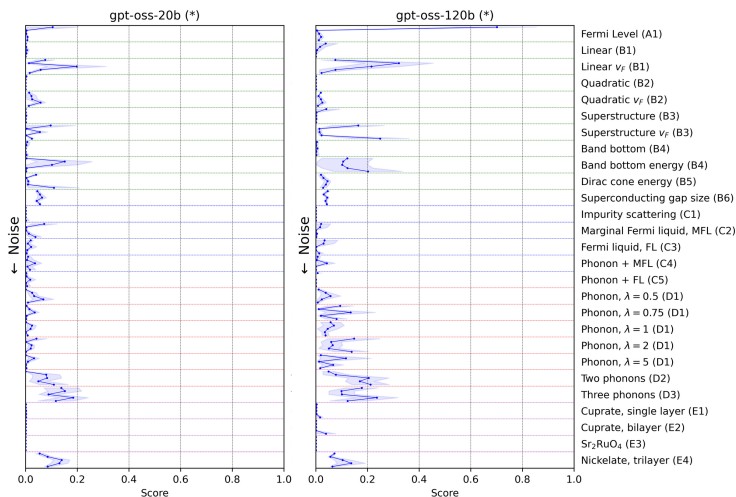

Figure 22: Scores from the open-weight models, gpt-oss-20b and gpt-oss-120b.

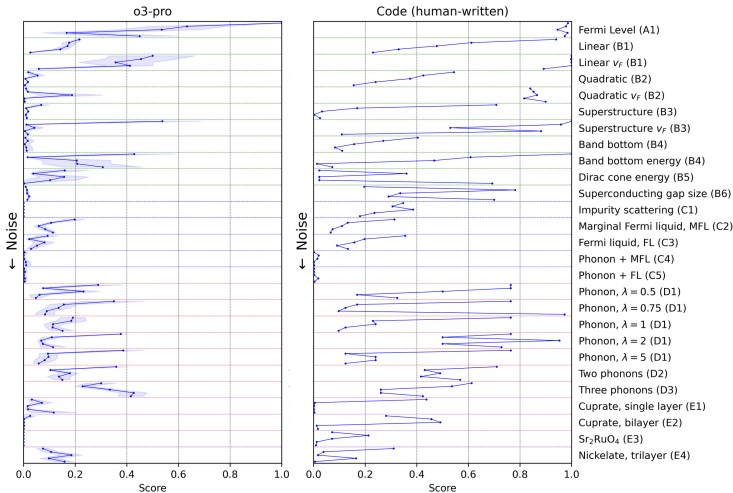

Figure 23: Scores from o3-pro and human–written code.

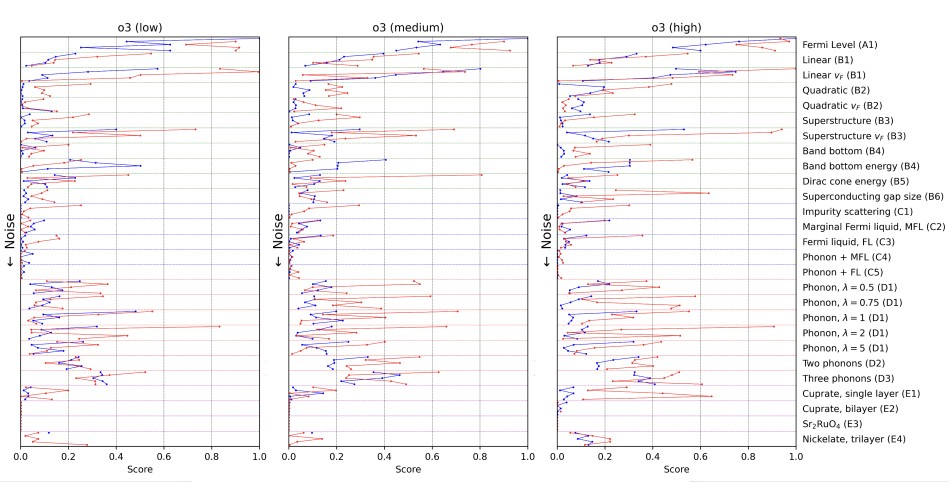

Figure 24: Scores from o3 for three inference–time compute modes *with* (red) and *without* (blue) code and tabulation enabled.

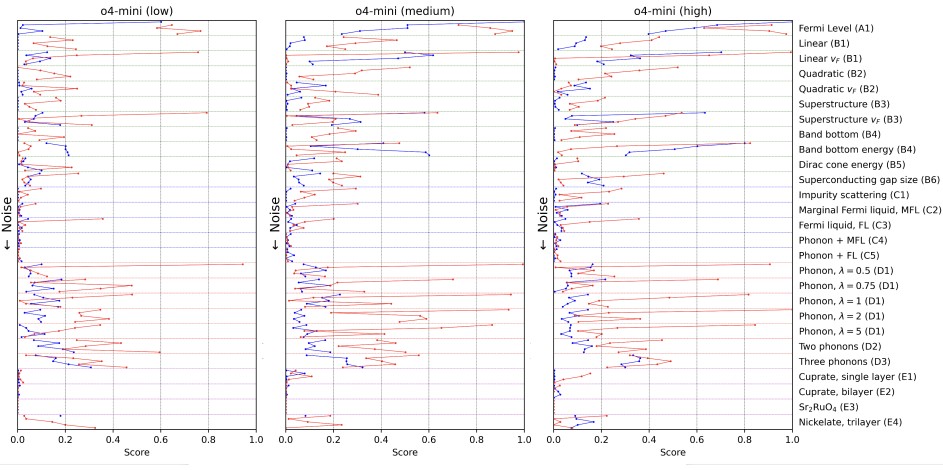

Figure 25: Scores from o4-mini for three inference–time compute modes *with* (red) and *without* (blue) code and tabulation enabled.

