# OpenReview forum: "SciPro Arena: a Case Study of AI Agent Capabilities in Scientific Analysis Tasks"
_ICLR.cc/2026/Conference — Submitted to ICLR 2026_

### Official Review · Reviewer_qLVe · 2025-10-28

**Soundness:** 3
**Presentation:** 4
**Contribution:** 3
**Rating:** 4
**Confidence:** 4

**Summary:**

This paper introduces a new benchmark, called SciPro Arena, to evaluate language models on real-world scientific analysis tasks. Given a stream of numerical data (e.g., a 2D intensity map with real energy and momentum axes) and some question, models extract patterns from examples and provide numerical answers which are scored by deviation from ground truth obtained from a realistic simulator. The authors evaluated 14 recent reasoning models both open- and closed-source on 135 questions (27 question types, each tested at 5 noise level times). Recent models tend to perform better (e.g., o3 and Gemini 2.5 Pro), but they only reach <15% on average while a human baseline program (~400 Python lines) scores 37%. Even when removing the noise, SOTA models score 20% while human program gets 55%. Since the questions are composed of different difficulty tiers, the authors could highlight that "[SOTA] models can extract simple features but fail at tracing continuous patterns or computing derived quantities, the latter constituting core reasoning skills needed for real scientific analysis".

**Strengths:**

- SciPro looks very challenging for current language models and according to the leaderboard it is far from saturation. Improving on that benchmark (esp. dealing with noise) will require breakthrough in reasoning/test-time compute.

- The benchmark offers different difficulty tiers which are great to track progress. The questions are grouped under 5 domains and seems to be easily extendable.

**Weaknesses:**

- To me it is unclear if directly evaluating models on numerical tasks totally makes sense. Specifically, what is the impact of the tokenization process on numerical values provided in the context. There's been some work in the literature on shedding light on typical error patterns of models on tasks involving numerical reasoning (e.g., https://arxiv.org/abs/2410.11781).

- For each question, three in-context examples are included as part of the prompt. According to the paragraph Form of questions in Section 3.2, each example contains a matrix of numbers with whitespace delimiters. While it provides some structured to the prompt, it is unclear to me what is the impact of presenting the information that way. To me in a realistic scenario the data would be ingested as a CSV file or some other structured format instead of free form text strings.

- The title of the paper contains "AI agent capabilities" but without tools or scaffolding that enable LLMs to interact with some environment, I wouldn't call the models being tested AI agents. Also, to be more realistic and comparable with the human baseline, I would argue the models should have been able to use tools, e.g. generating code, to deal with numerical values.

### Minor
Line 146 mentions that ARPES analysis is an inverse problem, but line 151 says "the aim of ARPES data analysis is then to work out x→ y". While I think I understand where that comes from, I find it more intuitive to think that the underlying dispersion and linewidth functions are the 'x' (of the forward process) and the noisy spectrum is the 'y', thus ARPES is really about solving y → x.

**Questions:**

- Did the authors try to equip the LLMs with tools (e.g., Python interpreter)? What about allowing them to write code? I suspect the performance will increase. If not, then that's a good motivation for SciPro. Also allowing the use of coding tool is more realistic for scientific analysis.

- Is providing 3 in-context examples optimal? Is it enough to capture the nature of the tasks or does the model still need prior knowledge about the scientific domain the question is coming from?

- I might have miss it but how are the answers extracted from the LLM's response?

- Will the benchmark be open-source?

---

> ### Author Response · Authors · 2025-11-21
>
> Weaknesses
> 1. We’ve added a new paragraph in Section 4.2 (Results) to address this.
> 2/3. We’re currently working on this, and we’ll keep you updated.
>
> Minor: this is a possible alternative way of framing the problem. We see your point too, though we have not yet come to a consensus on which is a more ‘natural’ phrasing. We’ll circle back to this!
>
> Questions
> 1/2. Same as 2/3 in weaknesses: we’ll keep you updated.
> 3.  We want only numerical values (possibly in array form, for array–like answers), which is enforced by structured output or running it through e.g. ChatGPT. In addition to this, we prompt models to return numbers or arrays.
> 4. Yes, it will be open–source.

---

> > ### Comment · Reviewer_qLVe · 2025-11-22
> >
> > Thanks for the update. The Tokenization dependence subsection looks good. It does seem to have some effects according to Fig 10 (interesting cyclic behavior depending on the tokenzation ratio) but I agree that this could be considered as another source of noise.

---

> > > ### Author Response · Authors · 2025-11-23
> > >
> > > Whether there is periodicity is a point we'll try to settle should we have the bandwidth: error bars for normalization of B1 are still relatively large (compared to A1, because of the relative difficulty of the question).
> > >
> > > (An aside. Incidentally, you've touched upon a point we've been mulling on, which we'll have to tackle in future work: the link between tokenization choices and information compression. Now, our original choice of normalizing intensities to a maximum of 1000 was a result of tests we ran, showing that earlier OpenAI models appeared to tokenize such that all integers 0—1000 were assigned a single token (strangely not 1024), such that this choice would maximize information under a naïve one–token–per–data point representation. We note that assigning one token to each number in a spectrum (= one pixel) means that only a single megapixel image would be able to fit into Gemini's 1 million token context window. Yet, we certainly know that images are tokenized completely differently, and in a far more efficient way, such that two orders of magnitude more megapixel images may be fit into the same context window. This points towards using far more efficient choices of representation than we had chosen — not necessarily as images, but possibly drawing upon lessons encountered by those who have worked on compressing them (FFTs, wavelets, and all that), or in constructing VLMs. Whether this will sacrifice the quantitative accuracy of our data is unclear, and so is dealing the bag of worms opened up by using VLMs and their attendant biases. The link between information compression and tokenization is a problem that will have to be dealt with pretty soon: loading larger scientific datasets, such as jets in high–energy physics (mentioned in the text), or even just 'stacks' of ARPES spectra (for 2D ARPES spectra are but mere slices of 3/4D bandstructure) will completely overwhelm current context windows. However, a choice of tokenization/compression is also a choice of representation that may interfere with either/both the fidelity of data contained, or/and a model's ability to quantitatively work with the data. It could be a subject of a future standalone paper. Just a few thoughts.)

---

> > > > ### Author Response · Authors · 2025-12-04
> > > > **Full comment**
> > > >
> > > > — Weakness 1 —
> > > >
> > > > Previously answered; see above.
> > > >
> > > > — Weakness 2 —
> > > >
> > > > See official comment.
> > > >
> > > > — Weakness 3 —
> > > >
> > > > We have edited the wording to de–emphasize agentic capabilities; also, see official comment about code.
> > > >
> > > > — Minor —
> > > >
> > > > We shall keep our original rephrasing: what we mean is that 'x' refers to information corrupted by convolution and noise (what the pixels show), whereas 'y' is the pure information we want to extract (dispersions and linewidths).
> > > >
> > > > — Question 1 —
> > > >
> > > > See official comment.
> > > >
> > > > — Question 2 —
> > > >
> > > > We have chosen three examples due to context window limits: increasing the number of examples either involves reducing the resolution size of each example, or dedicated pre–training/fine–tuning. We believe prior knowledge of the scientific domain is enough (i.e. basic information a sufficiently general model should have) to understand what the data means; the challenge is then to figure out how to get from the information contained in the raw data to what we want (dispersions, spectra). This is indeed the *central challenge* itself that models are tested on—the unelaborated inverse process—which we deliberately do not spoon–feed the models on. There is no one way to do this task, models are tested on their abilities to come up with a solution (by code or otherwise), and solutions may be arbitrarily complicated, e.g. Section 4.2: "However, the intrinsically ill–posed nature of inverse problems rears its head with higher noise, requiring increasingly sophisticated solutions to tame its difficulties, possibly indistillable into a compact coding problem, instead calling for advanced statistical treatment (Benning & Burger, 2018) or dedicated model training (Ying, 2022; Peng et al., 2020)."
> > > >
> > > > — Questions 3 & 4 —
> > > >
> > > > Previously answered; see above.

---

### Official Review · Reviewer_J31F · 2025-10-31

**Soundness:** 2
**Presentation:** 2
**Contribution:** 1
**Rating:** 2
**Confidence:** 3

**Summary:**

The authors propose a new benchmark, SciPro Arena, to test how well AI systems can analyze scientific data - specifically angle-resolved photoemission spectroscopy (ARPES) data. They test several frontier models, and find that in general models perform very poorly on the dataset, highlighting the continuing challenges of using AI for scientific discovery.

**Strengths:**

- Evaluation on a real scientific task in condensed matter physics
 - Reasonably rigorous scoring methodology

**Weaknesses:**

- While the abstract sets out to test "analysis of scientific data", the actual grounding of this on ARPES data seems very specific and idiosyncratic. ARPES has numerous complexities and specialities in, making it particularly difficult. If the authors goals are to more generally evaluate analysis of scientific data, they would perhaps do better first by characterizing different types/dimensions of scientific data, collecting samples of each, and doing a more systematic evaluation. In other words, the generalization from ARPES to all scientific data seems somewhat of a leap here.
 - Related to this, I'd really like to learn not how the models performs overall on this benchmark, but how the models perform on different aspects of scientific data analysis. Is it possible to identify different types of scientific reasoning required for this task? In other words, rather than (or as well as) having 27 domain-specific question types, identify N data analysis types (e.g., interpolation, prediction, pattern recognition, noise tolerance, data cleaning, visual interpretation, etc.). I'd love to see the paper's framing and conclusions mapped from the physics domain to the AI research domain more.
 - Frontier models appear to be applied in a vanilla/naive way, i.e., do a single-call direct query to the model. However, there are numerous "AI Scientist" systems out there that do coding, iterative reasoning, reflect loops (e.g., AIScientist, CodeScientist, ReAct, Reflexion, CodeAct) that might do better at this task. This should be clarified, in particular the conclusions may only apply to "direct query" uses of frontier models.
 - The results seem somewhat dependent on the prompting strategy used, e.g., choices of few-shot prompting
 - For several of the plots, I'm not sure what to take away from them - highlighting the takeaway in the caption, rather than just summarizing the visual data (e.g., "accuracy decrease with noise") would be very helpful.

**Questions:**

See weaknesses

---

> ### Author Response · Authors · 2025-11-21
>
> Weaknesses
> 1. We have amended this statement to be restricted to physics in particular, in which the same complications do recur, which we now address in point (3) in the second paragraph of the Introduction, as well as the end of section 3.1. In addition, I believe we did not properly address the significance of ARPES in the original version, a point which is now made in the Introduction.
> 2. We’ve overhauled the types of data analysis to reflect this (section 3.3); let us know if you agree with this, and if it needs more elaboration.
> 3/4. We are working on the tests to answer this, and we’ll keep you updated.
> 5. Good point; we’ll work on this when we’ve settled on the figures.

---

> > ### Author Response · Authors · 2025-11-23
> >
> > Here are the corresponding sections:
> >
> > 1. "(3) The core challenge of ARPES—finding patterns in noisy, multi–dimensional datasets—recurs across many subfields in physics where frontier AI systems will need to operate, and is thus a good proxy for other experimental techniques, particularly where spectra and images are analyzed. ARPES datasets are large enough to challenge context window limits of current models, while not being excessively high–dimensional where even rudimentary tasks would far exceed LLM limits, such as calorimetric data in high–energy physics (Baldi et al., 2016)."
> >
> > and: "The key attributes of ARPES data analysis complicating the solution of this inverse problem—that is, noise and convolution—recur for many modalities of data defined over continuous domains, which is the form that predominates in physics, and spectroscopic techniques in chemistry. In astronomical images, one encounters corruption by Poisson noise (Shamshad et al., 2018) convolved with point spread functions (instrumental resolution), such as the characteristic speckles of the James Webb Space Telescope (Kinakh et al., 2024). The study of jets in high–energy physics likewise involves problems of sifting through background noise (Sjölin, 2012) and calorimetric instrumental resolution Lobban et al. (2002)."
> >
> > 2. "Data analysis tasks    Each category involves at least one of four analytic tasks: regression (when simple mathematical formulae can be fit), structure determination (objects in spectra which are not straightforwardly mathematically described), noise dependence (all questions), and categorization of objects. A fuller explanation is given in Appendix D. Some tasks common in physics are not covered, such as anomaly detection and time series prediction."
> >
> > In appendix D: "Four data analysis tasks are identified. With the exception of noise dependence, scores for each task are calculated by averaging over the scores of questions tagged with each task.
> > • Regression. These apply when simple mathematical formula may in principle be fit onto spectra, which covers all questions except B5, B6, and the D of questions where the mathematical form is not apparent.
> > • Structure determination. This covers identifying such objects as superstructure in B3, B3 vF, band bottoms in B4 bbE, the Dirac cone in B5, separating the superconducting gap of B6 from the rest of the bandstructure, as well as phononic kinks in the D series and Fermi surfaces in the E series of questions.
> > • Categorization. These questions involve distinguishing between different objects in spectra and assign different values to them, such as the frequencies of multiple phononic kinks in D2 and D3, and the doping levels of separate bands in E2—4.
> > • Noise dependence. Scores are calculated within each question category by taking the ratio of the score coresponding to maximum noise over that of no noise, and subsequently averaged over all categories."

---

> ### Author Response · Authors · 2025-12-04
> **Full comment**
>
> — Weakness 1 —
>
> See official comment.
>
> Also, two major changes. (1) Restrict ourselves to the spectral/image analysis tasks in physics (and some of chemistry). From the Introduction: "The core challenges of ARPES (finding patterns in noisy, multi–dimensional datasets) recur across many subfields in physics, and is thus a good proxy for its other experimental techniques, particularly where spectra and images are analyzed. ARPES datasets are large enough to challenge context window limits of current models, ..."
>
> From Section 3.1: "The key attributes of ARPES data analysis complicating the solution of this inverse problem (that is, noise and convolution) recur for many modalities of data defined over continuous domains, the form that predominates in physics and chemical spectroscopies. In astronomical images, one encounters corruption by Poisson noise (Shamshad et al., 2018) convolved with point spread functions, such as the characteristic speckles of the James Webb Space Telescope (Kinakh et al., 2024). The study of jets in high–energy physics likewise involves problems of sifting through background noise (Sj¨ olin, 2012) and accounting for calorimetric instrumental resolution (Lobban et al., 2002)."
>
> (2) Rephrase these in more general terms. In Section 3.3: "Data analysis tasks Each category involves at least one of four analytic tasks: regression (when simple mathematical formulae can be fit), structure determination (objects in spectra which are not straightforwardly mathematically described), noise dependence (all questions), and categorization of objects. (Fuller explanations of these tasks are given in Appendix E.) Some tasks common in physics are not covered, such as anomaly detection and time series prediction."
>
> — Weakness 2 —
>
> We have made the following revision, including categorization of questions into data analysis tasks more familiar with the bulk of the AI community. See Section 3.3: "Data analysis tasks Each category involves at least one of four analytic tasks: regression (when simple mathematical formulae can be fit), structure determination (objects in spectra which are not straightforwardly mathematically described), noise dependence (all questions), and categorization of objects. (Fuller explanations of these tasks are given in Appendix E.) Some tasks common in physics are not covered, such as anomaly detection and time series prediction."
>
> And Section 4.2: "Comparing tasks A similar ordering of model performance was observed for the four data analytic tasks mentioned in Section 3.3, and presented in Figs. 7 and 8 in Appendix F, although a few surprises have showed up: Gemini 2.5 performed surprisingly poorly in tasks involving regression and categorization (relative to its overall performance), and Qwen3 found to be better for structure determination and categorization than other tasks."
>
> — Weaknesses 3 & 4 —
>
> See official comment.
>
> — Weakness 5 —
>
> We have updated captions to reflect this.

---

### Official Review · Reviewer_tPXF · 2025-11-02

**Soundness:** 3
**Presentation:** 2
**Contribution:** 2
**Rating:** 4
**Confidence:** 2

**Summary:**

This paper proposes a new benchmark called SciProArena that measures how good recent LLMs are for the task of scientific data analysis. In particular they focused on 27 categories of analysis tasks that require the models to extract patterns from noisy experimental data.  Authors present extensive experimental results covering recent reasoning models’ performance on the SciProArena benchmark by varying the noise level and dataset resolution.

**Strengths:**

* Unlike other benchmarks that focus on information extraction and inductive reasoning, SciPro Arena focuses on real, empirical data as input and requires deep analytical reasoning about complex, noisy data. The proposed task mirrors natural scientific experiments where relevant information (‘y’) is latent and must be inferred from proxy measurements (‘x’).

* Results demonstrate there is a large gap between human performance of 38 (55% on noiseless) vs most recent reasoning models achieve around 13% (21% on noiseless) on these tasks.

**Weaknesses:**

* Even though the dataset covers 27 task categories that cover A) Fermi level extraction, B) dispersion tracing, C) linewidth tracing, D) phonon energy determination, E) doping determination. While these five tasks are complex and represent core analytical work within their domain, they are a small fraction of the total landscape of scientific data analysis. Additional discussion about the scope of these tasks would help situate the claims better.

* As expected, the performance of reasoning models degrades as noise level or dataset resolution increases. More detailed discussion on what kind of agentic systems be developed on top of these LLMs to alleviate these limitations on existing tasks to solve the individual datapoints would strengthen the paper. Current future work focuses more on generalizing tasks or developing agents for meta-analysis.

**Questions:**

1. Placing result figures near their description will make the paper easy to read.

2. Authors have uploaded the supplementary material, however most of the prompt and result files are just placeholders. I would suggest releasing the prompts and data so that the research community can reproduce these results.

3. Section 3.4 explains how the noisy version of the dataset was generated by inserting randomly distributed 2D Gaussians in each spectrum.

---

> ### Author Response · Authors · 2025-11-21
>
> Weaknesses
> 1. You are right that they are not representative of much of scientific data analysis. We have amended this statement to be restricted to physics in particular, in which the same complications do recur, which we now address in point (3) in the second paragraph of the Introduction, as well as the end of section 3.1.
> 2. We are in the midst of answering this—we’ll update soon.
>
> Questions
> 1. Good point; we are still shifting figures around, but we’ll take it into account in later versions.
> 2. We’ll let you know when you’ve uploaded the files; we are still running additional tests.
> 3. We’ve updated the relevant explanation in the main text.

---

> > ### Author Response · Authors · 2025-11-23
> >
> > We quote the relevant section for the first weakness:
> >
> > "(3) The core challenge of ARPES—finding patterns in noisy, multi–dimensional datasets—recurs across many subfields in physics where frontier AI systems will need to operate, and is thus a good proxy for other experimental techniques, particularly where spectra and images are analyzed. ARPES datasets are large enough to challenge context window limits of current models, while not being excessively high–dimensional where even rudimentary tasks would far exceed LLM limits, such as calorimetric data in high–energy physics (Baldi et al., 2016)."
> >
> > and — "The key attributes of ARPES data analysis complicating the solution of this inverse problem—that is, noise and convolution—recur for many modalities of data defined over continuous domains, which is the form that predominates in physics, and spectroscopic techniques in chemistry. In astronomical images, one encounters corruption by Poisson noise (Shamshad et al., 2018) convolved with point spread functions (instrumental resolution), such as the characteristic speckles of the James Webb Space Telescope (Kinakh et al., 2024). The study of jets in high–energy physics likewise involves problems of sifting through background noise (Sjölin, 2012) and calorimetric instrumental resolution Lobban et al. (2002)."
> >
> > Regarding the addition of noise:
> > "Noise was inserted in the manner of Fig. 6(a) in (Kim et al., 2021): as a set of 2 × 105 randomly–distributed spots, whose intensities are randomly chosen within a range, and footprints broadened into 2D Gaussians."

---

> > > ### Comment · Reviewer_tPXF · 2025-11-27
> > >
> > > Thanks for responding to my concerns. I would encourage authors to make these revisions in the next draft of this paper.
> > > I will keep my review and scores.

---

> > > > ### Author Response · Authors · 2025-12-04
> > > > **Full comment**
> > > >
> > > > — Weakness 1 —
> > > >
> > > > We have done two things. The first is to restrict ourselves to the spectral/image analysis tasks in physics (and some of chemistry). From the Introduction: "The core challenges of ARPES (finding patterns in noisy, multi–dimensional datasets) recur across many subfields in physics, and is thus a good proxy for its other experimental techniques, particularly where spectra and images are analyzed. ARPES datasets are large enough to challenge context window limits of current models, ..."
> > > >
> > > > From Section 3.1: "The key attributes of ARPES data analysis complicating the solution of this inverse problem (that is, noise and convolution) recur for many modalities of data defined over continuous domains, the form that predominates in physics and chemical spectroscopies. In astronomical images, one encounters corruption by Poisson noise (Shamshad et al., 2018) convolved with point spread functions, such as the characteristic speckles of the James Webb Space Telescope (Kinakh et al., 2024). The study of jets in high–energy physics likewise involves problems of sifting through background noise (Sj¨ olin, 2012) and accounting for calorimetric instrumental resolution (Lobban et al., 2002)."
> > > >
> > > > The second is to rephrase these in more general terms. In Section 3.3: "Data analysis tasks Each category involves at least one of four analytic tasks: regression (when simple mathematical formulae can be fit), structure determination (objects in spectra which are not straightforwardly mathematically described), noise dependence (all questions), and categorization of objects. (Fuller explanations of these tasks are given in Appendix E.) Some tasks common in physics are not covered, such as anomaly detection and time series prediction."
> > > >
> > > > — Weakness 2 —
> > > >
> > > > We have added a comment in the Discussion section: "Firstly, agents should become fluent in the ‘domain language’ of any field to be able to represent a system in its full complexity and be poised to make profound inferences, compared to agents with a merely superficial grasp. Having found that code generation by agents is still insufficient, we anticipate fine–tuning models through RL as a next step, or giving agents access to train other (possibly simpler) models (Ying, 2022). In the future, we may need to construct domain–specific foundation models, or even find that transformer–based LLMs themselves are insufficient, calling for new architectures such as world models (LeCun, 2022)."
> > > >
> > > > — Question 1 —
> > > >
> > > > Good point. The caveat is that the camera–ready version is different from the double–blind version you're seeing, so there may be misalignment of figures. Nonetheless we have tried our best to accommodate this.
> > > >
> > > > — Question 2 —
> > > >
> > > > We have done so; the prompts are also in the supplementary material now.
> > > >
> > > > — Question 3 —
> > > >
> > > > The relevant section (3.4) has been updated: "Noise was inserted in the manner of Fig. 6(a) in Kim et al. (2021): as a set of 2 ×105 randomly–distributed spots, whose intensities are randomly chosen within a range, and footprints broadened into 2D Gaussians. The amount of noise is quantified as mean noise intensity as a fraction of maximum spectral intensity prior to adding noise. We do not measure against the mean spectral intensity, as this quantity varies with the size of the spectral feature investigated relative to the size of the whole dataset, whereas maximum spectral intensity does not; we are interested in the extraction of strong signals regardless of how much other information (the background signal) is present."

---

### Official Review · Reviewer_WySN · 2025-11-09

**Soundness:** 3
**Presentation:** 3
**Contribution:** 2
**Rating:** 4
**Confidence:** 4

**Summary:**

This paper introduces a physics-based benchmark for scientific data analysis. They focus on Angle–Resolved Photoemission Spectroscopy (ARPES) data, which often substitutes many condensed matter experiments and stands as a realistic benchmark for finding patterns in noisy, multidimensional datasets.

**Strengths:**

- I appreciate the authors' meticulous presentation of a domain that's complex to understand.
- Furthermore, it reaffirms that LLMs still struggle to perform scientific data analysis.
- The experiments are thorough and thoughtfully executed.
- I recommend that authors consider a physical sciences x AI-related workshop to publish this work. It's a great paper, but unfortunately, too narrow to publish in ICLR.

**Weaknesses:**

While I like authors' effort to carefully construct the benchmark, it still suffers from some issues:
- I like the focus on scientific data analysis, but this focus has been explored quite a bit over the last 1-2 years. For example, ScienceAgentBench, DiscoveryBench, and AutoDST are some prominent examples that focus on real scientific data analysis, in fact, expanding across multiple domains. All of them find similar results that LLM struggles to perform scientific data analysis requiring long-tail methods. The paper also missed these very relevant and important citations, while claiming "SciPro Arena fills a gap that has not been addressed before — analysis of real scientific data."
- To my earlier point, what additional insights this paper brings remain unclear. In other words, why AI model builders would test their models on this benchmark compared to earlier comprehensive ones, or why solving this benchmark dictates fundamental capabilities of an LLM, is unclear.
- "datasets within a question are contained in a single text string" -- does this mean to solve this benchmark, all you need is language-based reasoning? What if the data is presented in a tabular file, and the system can interact with the file using code (e.g., Python)? Why is the proposed setup more important than the latter?

**Questions:**

Please see the questions mentioned in the weakness.

---

> ### Author Response · Authors · 2025-11-21
>
> Weaknesses
> 1. We have modified the section on Related Work (section 2); see in particular the new penultimate paragraph. Let us know whether it answers your question.
> 2. The relevance of this benchmark rests firstly on its importance to physics, which in turn is a problem of representation of a physical system in its natural language (we have modified the second paragraph of the Introduction to reflect this). We could elaborate further on this point in the paper if you wish.
> 3. This is a great question; we are still running tests before we can state an answer in the paper. Stay tuned!

---

> > ### Author Response · Authors · 2025-11-23
> >
> > Also, we quote the corresponding sections for the points addressing weaknesses to be precise:
> >
> > 1. "The scientific process is an iterative cycle of interrelated tasks: hypothesis generation, experiment execution, data analysis, and communication. As the regular use of LLMs in science shifts beyond being merely an aid in brainstorming or writing (Liang et al., 2025), broad benchmarks have sprung up measuring their competence in other tasks, particularly by crystallizing these processes as code generation co–pilots (Chen et al., 2024) and workflow derivation (Majumder et al., 2024) problems. At the same time, efforts have been made towards automating the full research pipeline (Jansen et al., 2025; Lu et al., 2024; Yamada et al., 2025). The question of whether data analysis can be tackled has previously been treated by Liu et al. (2024b) on far smaller, non–scientific datasets. Now, it has to be noted that the role of the human in the process of scientific discovery may be thought of as an act of ‘controlled rebellion’ (Polanyi, 1962): the originality (degree of surprise) of their discovery held in tension against high standards of proof enforced by the inherently conservative attitude of science towards modifying consensus, standards judged and upheld by other human scientists. Any agent seeking to supplement, or supplant, the work of the human in science must match the same high bar, or find its results disregarded by the scientific community. Our benchmark is thus targeted at this weak link in the scientific process: the question of whether models can be trusted on data analysis, emphasizing the maintenance of full standards of rigor, accountability, and interpretability."
> >
> > 2. Particularly point 2 — "We choose ARPES for three reasons. (1) Condensed matter shares with AI the common thrust (Xiao et al., 2025) of studying extremely complex systems (on the order of Avogadro’s constant, ∼ 1023). However, it lacks the large number of analyzable features afforded to LLMs (viz. Golden Gate Claude (Templeton et al., 2024)) because these features are accessed only if an experimental technique is physically feasible. Of the relatively limited number of techniques available, ARPES stands out for its relatively rich and less severely collapsed feature space, and closeness to the ground truth (more direct linkage to theoretical models), requiring less theoretical scaffolding for its interpretation. (2) Domain–specific foundation models elsewhere in science (Nguyen et al., 2024; Kim et al., 2021; Chithrananda et al., 2020) have emphasized the importance of learning the ‘domain language’ in which scientific data is naturally represented (Zhang et al., 2025). In condensed matter, embodying an understanding of electronic structure (grokking ARPES data)—its ‘domain language’—is a necessary condition for learning an informative representation of material systems. (3) The core challenge of ARPES—finding patterns in noisy, multi–dimensional datasets—recurs across many subfields in physics where frontier AI systems will need to operate, and is thus a good proxy for other experimental techniques, particularly where spectra and images are analyzed. ARPES datasets are large enough to challenge context window limits of current models, while not being excessively high dimensional where even rudimentary tasks would far exceed LLM limits, such as calorimetric data in high–energy physics (Baldi et al., 2016)."

---

> ### Author Response · Authors · 2025-12-04
> **Full comment**
>
> Revisions of our answers:
>
> — Weakness 1—
>
> We have restricted our domain to spectra and images, which are important in Physics (and parts of chemistry. The references are relevant (although AutoDST is a database rather than a benchmark). We have taken note of previous work in scientific benchmarks, and relegated our previous paragraphs on older benchmarks to an appendix (Appendix A). We have also made some modifications to reflect this.
>
> From the Introduction: "On the cusp of widespread adoption of AI systems in science, we need tests for rigorous, dependable scientific reasoning. Recent scientific benchmarks test competence in code and/or workflow generation..."
>
> In Related Work: "Scientific benchmarks acknowledge that the scientific process is an iterative cycle of interrelated tasks—hypothesis generation, experiment execution, data analysis, and communication—and correspondingly test either specific tasks in that cycle, or the whole process itself. Regular scientific use of LLMs has now shifted beyond mere brainstorming/writing aids (Liang et al., 2025), and recent benchmarks have sprung up measuring the competence of models in some of these tasks, particularly by crystallizing them as code generation (Chen et al., 2024) and workflow derivation problems (Majumder et al., 2024). Efforts have been made towards automating the full research pipeline (Jansen et al., 2025; Lu et al., 2024; Yamada et al., 2025). The question of whether data analysis can be tackled had been treated by Liu et al. (2024b) on smaller, non–scientific datasets."
>
> — Weakness 2 —
>
> Our reply comes in two portions, which we quote here.
>
> Why test on our benchmark, rather than earlier comprehensive ones; see Related Work: "recent benchmarks have sprung up measuring the competence of models in some of these tasks, particularly by crystallizing them as code generation (Chen et al., 2024) and workflow derivation problems (Majumder et al., 2024). Efforts have been made towards automating the full research pipeline (Jansen et al., 2025; Lu et al., 2024; Yamada et al., 2025). [...] SciPro Arena is targeted at the weakest link in the scientific process: it asks whether models can be trusted on data analysis, emphasizing maintenance of full standards of rigor, accountability, and interpretability. It is crucial to note that the role of an agent or human in scientific discovery lies in the act of ‘controlled rebellion’ (Polanyi, 1962), the originality or degree of surprise of their discovery held in tension against high standards of proof enforced by the inherently conservative attitude of science towards modifying consensus — standards judged and upheld by other human scientists. Any agent seeking to supplement, let alone supplant, the work of a human in science must first match the same high bar, or find its results disregarded by the scientific community."
>
> Fundamental limits of LLMs: the summary is that models can only score well if they are able to learn the domain language of a field, which may not be tokenized in a compact or optimal way, stretching context window limits. I.e. the best representation of the information relevant to understanding a complex system is not necessarily the way data is represented, so models must be able to bridge this divide.
>
> See point 2 in the Introduction: "Domain–specific foundation models elsewhere in science have emphasized the importance of learning the ‘domain language’ in which scientific data is naturally represented (Zhang et al., 2025), such as genomes in biology (Nguyen et al., 2024), and molecules in chemistry (Chithrananda et al., 2020; Kim et al., 2021). Embodying an understanding of electronic structure, the ‘domain language’ of condensed matter, is a necessary condition for learning an informative representation of material systems."
>
> Also see the following paragraphs in the Discussion: "We anticipate that several developments are necessary before the full potential of agents is harnessed in condensed matter. Firstly, agents should become fluent in the ‘domain language’ of any field to be able to represent a system in its full complexity and be poised to make profound inferences, compared to agents with a merely superficial grasp. Having found that code generation by agents is still insufficient, we anticipate fine–tuning models through RL as a next step, or giving agents access to train other (possibly simpler) models (Ying, 2022). In the future, we may need to construct domain–specific foundation models, or even find that transformer–based LLMs themselves are insufficient, calling for new architectures such as world models (LeCun, 2022)."
>
> — Weakness 3 —
>
> See official comment.

---

### Author Response · Authors · 2025-12-04
**Official comment**

In individual responses we have sometimes written "see official comment", which refers to this. These reflect two broad comments raised by the majority of reviewers, which we shall address in one common response here.

(1) Significance of the benchmark, why it matters, and why ARPES matters.

The importance of the benchmark encompasses two reasons. Firstly, it generally tests whether models are able to understand the intrinsic representation of the data of a field in its natural language, i.e. the domain language, rather than in a pre–processed form. Secondly, and more specifically, the importance of ARPES, which we feel we had downplayed in earlier drafts. It is now made explicit.

In the Introduction: "Condensed matter, the field responsible for developing much of modern technology—especially computing hardware that led to the advent of AI itself—promises scientific advances that will beget further breakthroughs in computing and other sciences (de Leon et al., 2021). Electronic structure is its ‘genomic code’, telling us how electrons behave in materials, thereby explaining material properties. Condensed matter shares with AI the common thrust (Xiao et al., 2025) of studying extremely complex systems (on the order of Avogadro’s constant,∼1023), but lacks the generous breadth of analyzable features afforded to LLMs, viz. Golden Gate Claude (Templeton et al., 2024), because these features are accessible only if an experimental technique is physically feasible."

"We focus on the tip of the spear of condensed matter, Angle–Resolved Photoemission Spectroscopy (ARPES) (Damascelli et al., 2003; Sobota et al., 2021), which measures electronic structure. ARPES was chosen for three strategic reasons: [...] Domain–specific foundation models elsewhere in science have emphasized the importance of learning the ‘domain language’ in which scientific data is naturally represented (Zhang et al., 2025), such as genomes in biology (Nguyen et al., 2024), and molecules in chemistry (Chithrananda et al., 2020; Kim et al., 2021). Embodying an understanding of electronic structure, the ‘domain language’ of condensed matter, is a necessary condition for learning an informative representation of material systems. Understanding electronic structure is crucial (Goyal et al., 2025) for constructing foundation models for the critical field of novel materials, the reason for which condensed matter is all–important
(Trump, 2025). Because grokking ARPES data is the only way scientists currently have to reveal electronic structure (Yang et al., 2018), the ability to process ARPES data is essential to leveraging the capabilities of frontier models to contribute decisively to this sector of technological advancement."

Additionally, have overhauled the section on Related Work: "SciPro Arena is targeted at the weakest link in the scientific process: it asks whether models can be trusted on data analysis, emphasizing maintenance of full standards of rigor, accountability, and interpretability. It is crucial to note that the role of an agent or human in scientific discovery lies in the act of ‘controlled rebellion’ (Polanyi, 1962), the originality or degree of surprise of their discovery held in tension against high standards of proof enforced by the inherently conservative attitude of science towards modifying consensus — standards judged and upheld by other human scientists. Any agent seeking to supplement, let alone supplant, the work of a human in science must first match the same high bar, or find its results disregarded by the scientific community."

(cont.)

---

### Author Response · Authors · 2025-12-04
**Official comment (cont.)**

(2) Prompting: code and tabulation instead of a single text string.

We have done what the majority of you have asked, and tested two of the best models (o3 and o4–mini) on three compute modes each with prompts containing data tabulated in .csv rather than stated in .txt, as well as given access to a Python interpreter, and have seen improvements, particularly to noiseless questions, but less so in question with strong noise.

Numerous changes were made at various points to reflect this result (see Abstract, Introduction, Fig. 1, Conclusion) and a new section in section 4.2 (Results): "Python interpreter and tabulation Access to a Python interpreter was additionally given to o3 and o4–mini on all three compute modes. Tabular data was presented in .csv and models were prompted to produce code which was in turn input to the interpreter, whose numerical output were scored (Figs. 1 inset, 24, and 25). Gains in score were substantial at low noise, but marginal with severe noise. This may reflect the corresponding increase in difficulty in mapping inverse problems onto code generation problems. At zero noise, an analytic solution may exist and is conceivably retrievable by simple code. However, the intrinsically ill–posed nature of inverse problems rears its head with higher noise, requiring increasingly sophisticated solutions to tame its difficulties, possibly indistillable into a compact coding problem, instead calling for advanced statistical treatment (Benning & Burger, 2018) or dedicated model training (Ying, 2022; Peng et al., 2020)."

---

### Meta-Review · Area_Chair_RL6W · 2025-12-09

**Summary:**

The reviewers-- unanimously-- recommended not accepting the papers, with scores between 2 and 4. After reading the reviews, your responses, and the paper itself, I am following their recommendation.

I do think that the paper is addressing a highly important and interesting problem, and that the benchmark and empirical results are valuable and seem sound. However, I do share the reviewer's concern that a major problem with the paper is the framing of the contribution, including the title. The paper is about a _particular_ set of analyses in a _particular_ domain (ARPES), but represented as a general 'arena' of 'scientific analysis' tasks-- this is an unfortunate misrepresentation of the (very good and valuable, and focused) contribution. The authors have very good arguments why this is a great 'model system' to study scientific analyses, but it would still be important that the paper is clearly presented.

I am convinced that this can---and will---be a great paper and highly useful contribution. In its current form, I do think it is not yet, and it was also my assessment that the chances for the final version would be too substantial to allow accepting it based on the information currently available.

**Reviewer Concerns:**

see meta review and their points.

**Reviewer Scores:**

I am unwilling to follow this request. The ACs are being asked to perform an unreasonable amount of extra work this year. In particular, I understand the concern about potential collusion. However, I do think that asking ACs to now read a whole new set of papers, reviews and (extremely long...) rebuttals and to try to condense them into decisions without a chance for discussions is both asking a lot from us, and will, inevitably, result in (on average) poorer decisions for everyone.

I do not see any value in spending even more time to try to 'guess' how each reviewer would have changed their mind or not. I am just trying to make appropriate decisions on the paper and focus on the science. The authors can see the reviewer inputs and should take it into account.

---

### Decision · Program_Chairs · 2026-01-26

Reject